# Metabolic stress-induced cardiomyopathy is caused by mitochondrial dysfunction due to attenuated Erk5 signaling

Wei Liu[1], Andrea Ruiz-Velasco [1], Shoubao Wang[1,2], Saba Khan[1], Min Zi[1], Andreas Jungmann[3], Maria Dolores Camacho-Muñoz[1], Jing Guo[1], Guanhua Du[2], Liping Xie[4], Delvac Oceandy [1], Anna Nicolaou[1], Gina Galli[1], Oliver J. Müller [3,5], Elizabeth J. Cartwright[1], Yong Ji[4] & Xin Wang[1]

The prevalence of cardiomyopathy from metabolic stress has increased dramatically; however, its molecular mechanisms remain elusive. Here, we show that extracellular signal-regulated protein kinase 5 (Erk5) is lost in the hearts of obese/diabetic animal models and that cardiac-specific deletion of *Erk5* in mice (Erk5-CKO) leads to dampened cardiac contractility and mitochondrial abnormalities with repressed fuel oxidation and oxidative damage upon high fat diet (HFD). Erk5 regulation of peroxisome proliferator-activated receptor γ co-activator-1α (Pgc-1α) is critical for cardiac mitochondrial functions. More specifically, we show that Gp91phox activation of calpain-1 degrades Erk5 in free fatty acid (FFA)-stressed cardiomyocytes, whereas the prevention of Erk5 loss by blocking Gp91phox or calpain-1 rescues mitochondrial functions. Similarly, adeno-associated virus 9 (AAV9)-mediated restoration of Erk5 expression in Erk5-CKO hearts prevents cardiomyopathy. These findings suggest that maintaining Erk5 integrity has therapeutic potential for treating metabolic stress-induced cardiomyopathy.

[1] Faculty of Biology, Medicine and Health, The University of Manchester, Michael Smith Building, Oxford Road, Manchester M13 9PT, UK. [2] Institute of Materia Medica, Chinese Academy of Medical Sciences & Peking Union Medical College, Nanwei Road, Xuanwu District, Beijing 100050, China. [3] Internal Medicine III, University Hospital Heidelberg, Im Neuenheimer Feld 410, 69120 Heidelberg, Germany. [4] Key laboratory of Cardiovascular & Cerebrovascular Medicine, School of Pharmacy, Nanjing Medical University, 101 Longmian Road, Nanjing, Jiangsu 211166, China. [5] Department of Internal Medicine III, University of Kiel, Schittenhelmstrasse 12, 24105 Kiel, Germany. Wei Liu, Andrea Ruiz-Velasco, Shoubao Wang and Xin Wang contributed equally to this work. Correspondence and requests for materials should be addressed to W.L. (email: wei.liu@manchester.ac.uk) or to Y.J. (email: yong.ji@njmu.edu.cn) or to X.W. (email: xin.wang@manchester.ac.uk)

W e are witnessing a rapid epidemic-like rise in the prevalence of obesity, which is a major driving force for the development of type 2 diabetes (T2D). It is estimated that globally more than 1.1 billion adults are overweight, of whom ~312 million are obese. With a rise in obesity on such a large scale, diabetes affects nearly 200 million people around the world and by 2025 it is expected that this number will reach to 333 million[1]. Compared with age-matched non-diabetics, diabetic patients have a nearly threefold increase in the risk of developing heart failure[2], which progresses from metabolic cardiomyopathy, a collective condition featured by insulin resistance, suppressed fuel oxidation, ROS accumulation, apoptosis and deterioration in contractility[3, 4]. Mitochondrial dysfunction is proposed to underlie these abnormalities.

Mitochondria are the primary energy-generating organelles in the heart. The immense energy demands of the working heart are largely satisfied by ATP produced via mitochondrial oxidative phosphorylation (OXPHOS), from which fatty acid oxidation (FAO) contributes ~70% of the energy supply, whereas the rest is met by glucose oxidation[5]. Acting as the primary ATP suppliers and oxygen consumers, mitochondria exert their functions under exquisitely controlled gene regulation circuits, which work through a set of the nuclear receptor superfamily, including the peroxisome proliferator-activated receptors (Ppars), estrogen-related receptors (Errs) and nuclear respiratory factors (Nrfs), and Pparγ co-activator 1 (Pgc-1α and Pgc-1β)[6]. Experimental and clinical studies have shown that in the adaptive stage of metabolic stress-induced cardiomyopathy, the activity of both Pgc-1α and Pparα is high, which drives the balance of energy metabolism toward greater fatty acid oxidation to generate more ATP. However, over time, an unremitting metabolic stress will lead to downregulation of Pgc-1α and its downstream cascades with repression of both FAO and glucose utilization, therefore resulting in cardiac lipid overload, insulin resistance, energy deficiency, increased ROS production and apoptosis, finally leading to heart failure[7]. A key question stemming from this pathological process is that the mechanism whereby Pgc-1α levels fall remains largely unknown, precluding the development of upstream therapeutic strategies.

Erk5 is an atypical mitogen-activated protein kinase with transcriptional activity. Its C-terminal tail contains a trans-activation domain for regulating a number of transcription factors, such as myocyte enhancer factor 2 (Mef2), kruppel-like factor 2/4 (Klf2/4) and cAMP response element-binding protein (Creb)[8, 9]. Through these transcription factors, Erk5 participates in a host of biological events, including cell proliferation, survival, muscle maturation, and inflammatory response[8–11].

In this study, we first discovered that Erk5 was selectively degraded in the myocardium of several obese/diabetic animal models. Phenotypic analysis demonstrated that 16-week high fat diet (HFD) feeding caused heart dysfunction in Erk5-CKO mice. The Erk5-CKO hearts displayed a spectrum of marked mitochondrial abnormalities with downregulation of Pgc-1α and its downstream genes. Concurrently, fatty acid overload, ROS accumulation and apoptosis were apparent in Erk5-CKO myocardium. Mechanistic studies revealed that FFA-caused Erk5

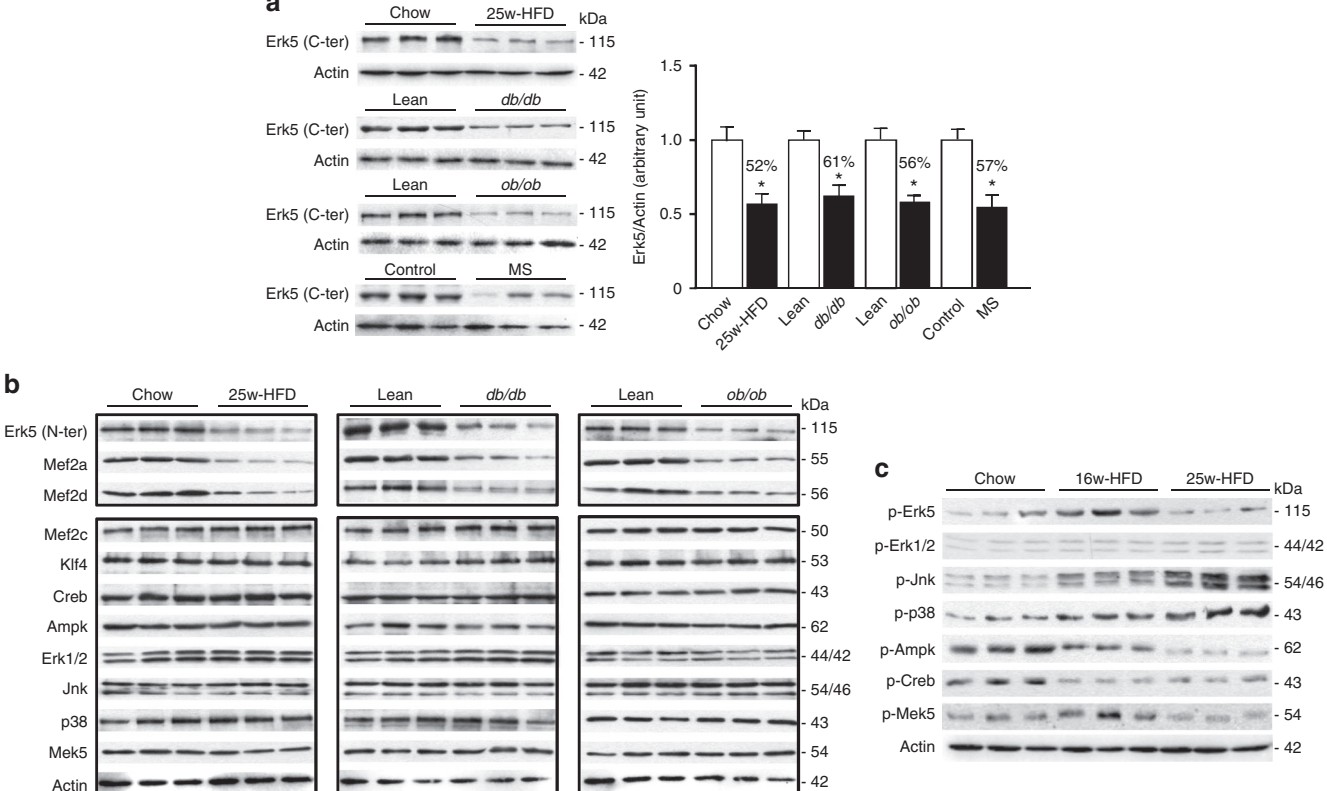

**Fig. 1** Decreased expression and phosphorylation of Erk5 in the obesity/diabetic hearts. **a** Immunoblot analyses (antibody recognizing C-terminus of Erk5) showed a decrease of Erk5 expression in the hearts from C57BL/6J mice with 25-week HFD, db/db mice, ob/ob mice, or rhesus monkeys with metabolic syndrome (MS). Quantification of Erk5/Actin ratio is presented in the bar graph. Data are means ± SD (*P < 0.05, vs. control groups, n = 6 animals per group). **b** Immunoblot analyses substantiated that expression level of Erk5 (antibody recognizing N-terminus of Erk5), Mef2a and Mef2d was decreased in the obesity/diabetic hearts. **c** The phosphorylation levels of Erk5, Erk1/2, Jnk, p38, Ampk, Creb, and Mek5 in the hearts from mice with 16-week or 25-week HFD were examined by immunoblot analyses. Actin is the protein loading control

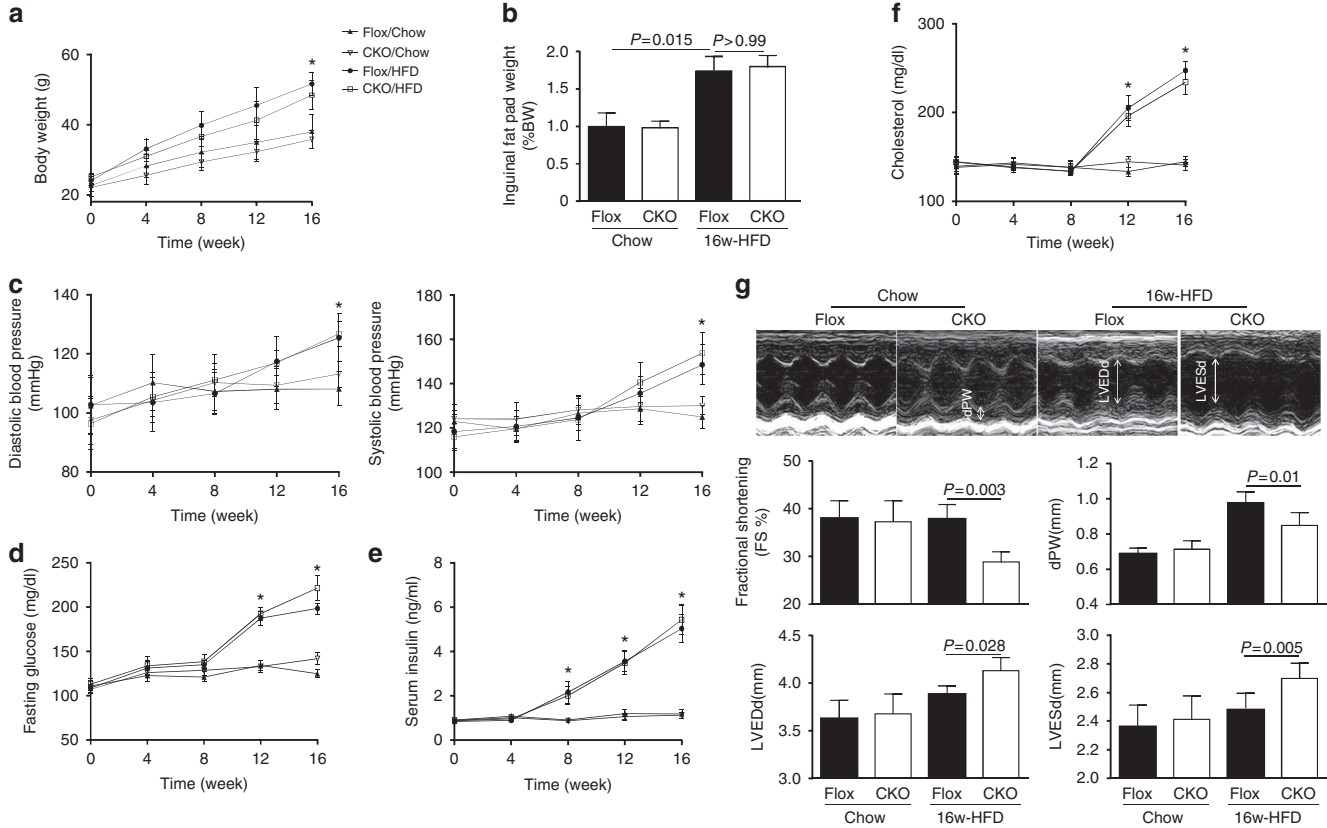

**Fig. 2** Impaired cardiac function in CKO mice with 16-week HFD. **a** Body weights of Flox and CKO mice fed with chow diet or HFD were recorded at 4-week intervals. (*$P < 0.05$, vs. chow groups, $n = 11$ mice per group). **b** Quantitative analysis of inguinal fat-pad mass. ($n = 6$ mice per group). **c** Diastolic and systolic blood pressure, **d** fasting blood glucose, **e** insulin, and **f** cholesterol were gradually increased with HFD for 16 weeks in both Flox and CKO mice. *$P < 0.05$, vs. chow groups, $n = 9$ mice per group. **g** Echocardiographic assessment of FS%, dPW, LVEDd, and LVESd by M-mode images and quantification demonstrated the impaired cardiac function in HFD-CKO mice. $n = 12$ mice per group. Data are presented as means ± SD

degradation was dependent on Gp91phox (Nadph oxidase) activation of calpain-1. Knockdown of Gp91phox, calpain-1 or inhibiting its activity could restore Erk5 expression, which replenished Pgc-1α expression and rescued mitochondrial functions. Consistently, AAV9-based re-establishment of Erk5 in Erk5-CKO hearts was able to prevent HFD-induced cardiomyopathy. In addition, our molecular data show that Erk5 regulation of Mef2 is responsible for Pgc-1α transcriptional expression. Collectively, these data illustrate a new protective mechanism for proper mitochondrial functions by Erk5 positive regulation of Pgc-1α, thus giving solid credence to the possibility of sustaining Erk5 integrity as a therapeutic approach for treating metabolic stress-induced cardiomyopathy.

## Results

**Erk5 selective reduction in various obese/diabetic hearts.** To evaluate Erk5 involvement in response to metabolic stress from obesity and diabetes, we first examined Erk5 expression in various obese and diabetic animal models, such as C57BL/6J mice fed with 25-week HFD, two diabetic mouse models of db/db and ob/ob mice, and Rhesus monkeys with spontaneous metabolic syndrome[12]. Interestingly, we disclosed the unique protein expression pattern that Erk5 expression was remarkably decreased in the hearts of all examined animal models, but not in their livers or skeletal muscles, which are also mitochondria-enriched organs (Fig. 1a, b; Supplementary Fig. 1a, b). Two different antibodies recognizing Erk5 N-terminus and C-terminus, respectively, confirmed Erk5 reduction (Fig. 1b),

whereas its mRNA level was not changed (Supplementary Fig. 2). In association with Erk5, we also found reduced expression of Mef2a and Mef2d (Fig. 1b), whereas protein expression remain unchanged for Mef2c, Klf4, Creb, Mek5, Erk1/2, p38, Jnk, and Ampk in the hearts, livers and skeletal muscles of obese mice (Fig. 1b; Supplementary Fig. 1a, b). In parallel, we also screened phosphorylation of Erk5, p38, Jnk, Ampk, Creb, and Mek5, we found that phosphorylation levels of Erk5, p38, Jnk, and Mek5 were increased in the hearts of C57BL/6J mice at the time of 16-week HFD; however, phosphorylation of Erk5 and Mek5 was considerably reduced at the time of 25-week HFD, whereas phosphorylation of p38 and Jnk remained higher. Of note, Ampk and Creb phosphorylation levels were gradually decreased responsive from 16-week to 25-week HFD feeding, whereas Erk1/2 phosphorylation was not changed (Fig. 1c). Given the reduction in both expression and phosphorylation of Erk5, an involvement of Erk5 in cardiac pathology of obesity and diabetes is obviously suggested.

**Cardiac Erk5-deficient mice display impaired contractility.** Hyperlipidemia and hyperglycemia from obesity/diabetes damage the heart leading to a unique disease condition known as metabolic stress-induced cardiomyopathy. To investigate a direct relationship between Erk5 deficiency and this cardiomyopathy, we used Erk5-CKO mice to investigate the role of Erk5 in the obese/diabetic hearts. Erk5-CKO and control mice (Erk5-Flox) were fed with HFD for 16 weeks. Of note, at this time point Erk5 expression was maintained in Erk5-Flox hearts (Supplementary

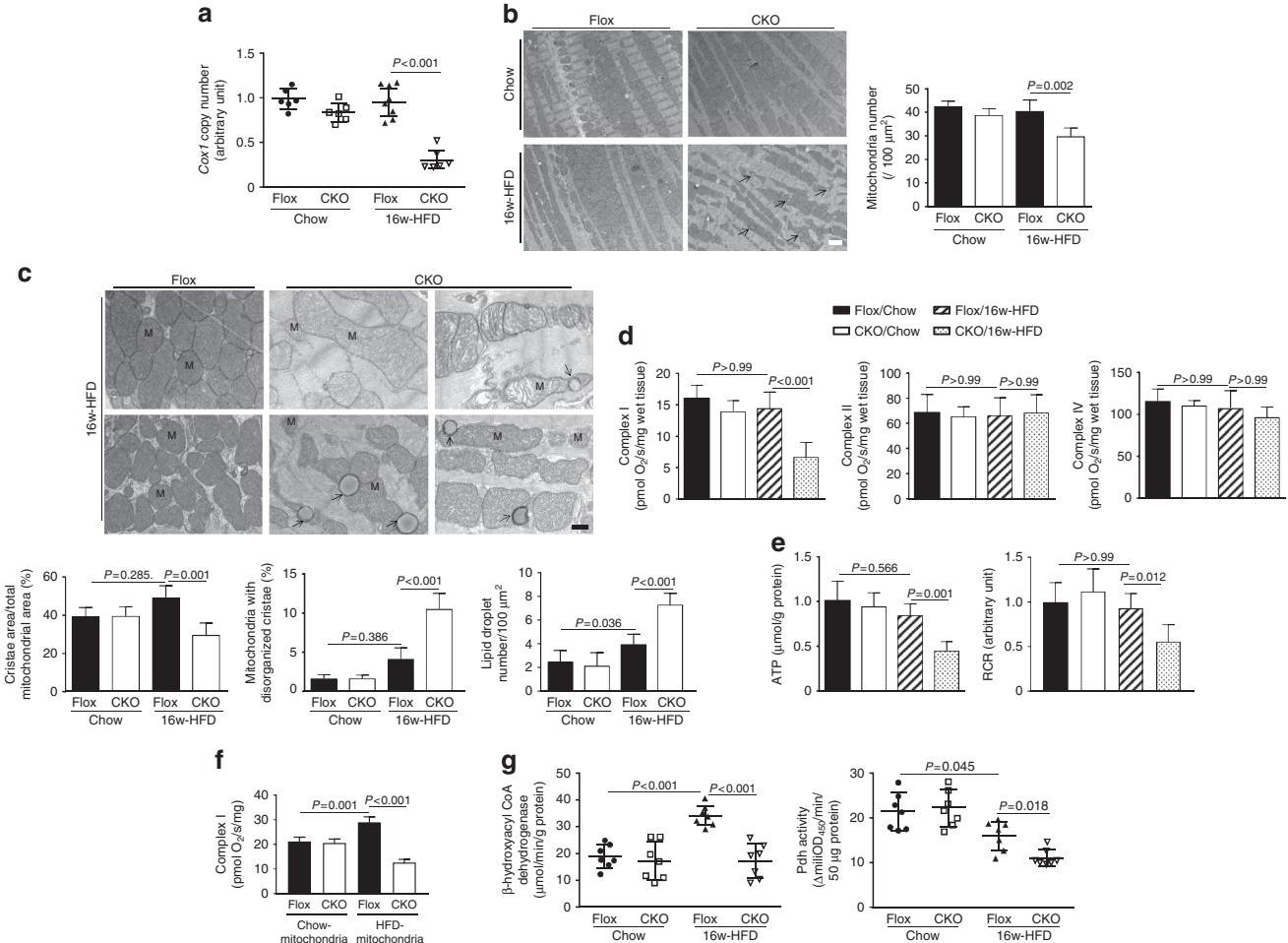

**Fig. 3** Mitochondrial morphological and functional aberrations in HFD-CKO hearts. **a** The content of mtDNA was measured by the ratio of Cox1 to cyclophilin A. **b** Mitochondria was examined by TEM (×1400). Representative images showing reduced and disorganized mitochondria cluttering in the cytosol of HFD-CKO hearts (*scale bar*: 2 μm). The quantitative analysis of mitochondria number is provided in bar graphs. **c** Higher magnification (×4800) of TEM images show disarrayed cristae, vacuoles and a reduced electron density of the matrix in the HFD-CKO mitochondria (*scale bar*: 0.5 μm), M symbolize mitochondria, arrows indicate lipid droplets. Quantitative analysis of cristae area, disorganized cristae in mitochondria, lipid droplet number, respectively, is provided in the bar graphs ($n = 6$ mice per group). **d** Respiratory rates of mitochondrial ETC complexes were measured in saponin-permeabilized myocardial fibers. With glutamate/malate as substrates, complex I respiratory rate was found to be decreased in HFD-CKO hearts. **e** Significantly reduced ATP production in HFD-CKO hearts was determined. Consistent with this, the respiratory control ratio (RCR) was lower in HFD-CKO hearts. **f** Analysis of complex I in mitochondria homogenates using palmitoylcarnitine as a substrate showed that ERK5-deficient mitochondria of HFD-fed mice had a reduction in FA-driven respiration. **g** Mitochondrial fuel oxidation was suppressed in HFD-CKO hearts evidenced by activity measurement of β-hydroxylacyl CoA dehydrogenase and pyruvate dehydrogenase (Pdh) ($n = 7$ mice per group). Data are presented as means ± SD

Fig. 1c). Phenotypic analysis showed that nesting time was comparable among experimental groups, whereas food intake was higher attendant with body weight, inguinal fat-pad mass and blood pressure being elevated in HFD-fed groups compared with chow-fed groups (Fig. 2a–c; Supplementary Fig. 3). Additionally, metabolic profiles measured by fasting glucose, serum insulin and circulating cholesterol demonstrated increased levels in HFD-fed groups (Fig. 2d–f). Hypertrophic growth is a pathological response of cardiac myocytes to HFD. Morphological analysis showed that the increased cross-sectional area of cardiomyocytes by 16-week HFD feeding was larger in Flox hearts than in CKO hearts, whereas interstitial fibrosis was more apparent in HFD-CKO hearts (Supplementary Fig. 4). Cardiac function was assessed at 4-week intervals by echocardiography showing that baseline measurements were similar between CKO and Flox groups. However, the cardiac contractile function was notably reduced in HFD-CKO mice after 16-week feeding, exemplified by decreased fractional shorting (FS). Decreased end-diastolic

posterior wall thickness (dPW), increased left ventricular end-diastolic diameter (LVEDd) and left ventricular end-systolic diameter (LVESd) were detected (Fig. 2g; Supplementary Fig. 5). These results demonstrate obesity establishment by HFD in both experimental models, and that hearts defective of Erk5 have less hypertrophy and blunted capacity to withstand HFD stress.

**Erk5 deficiency induces mitochondrial aberrations.** Impaired contractility correlates with reduced ATP synthesis and mitochondrial dysfunction. Therefore, we examined mitochondrial morphology and functions. First, we investigated the mitochondrial density in HFD-CKO and HFD-Flox hearts. As shown in Fig. 3a and Supplementary Fig. 6, the ratio of mitochondrial DNA (mtDNA) to nuclear DNA declined in the hearts of Erk5-CKO mice along with HFD feeding duration. Furthermore, the transmission electron microscopy (TEM) revealed that HFD-CKO hearts had disorganized mitochondria

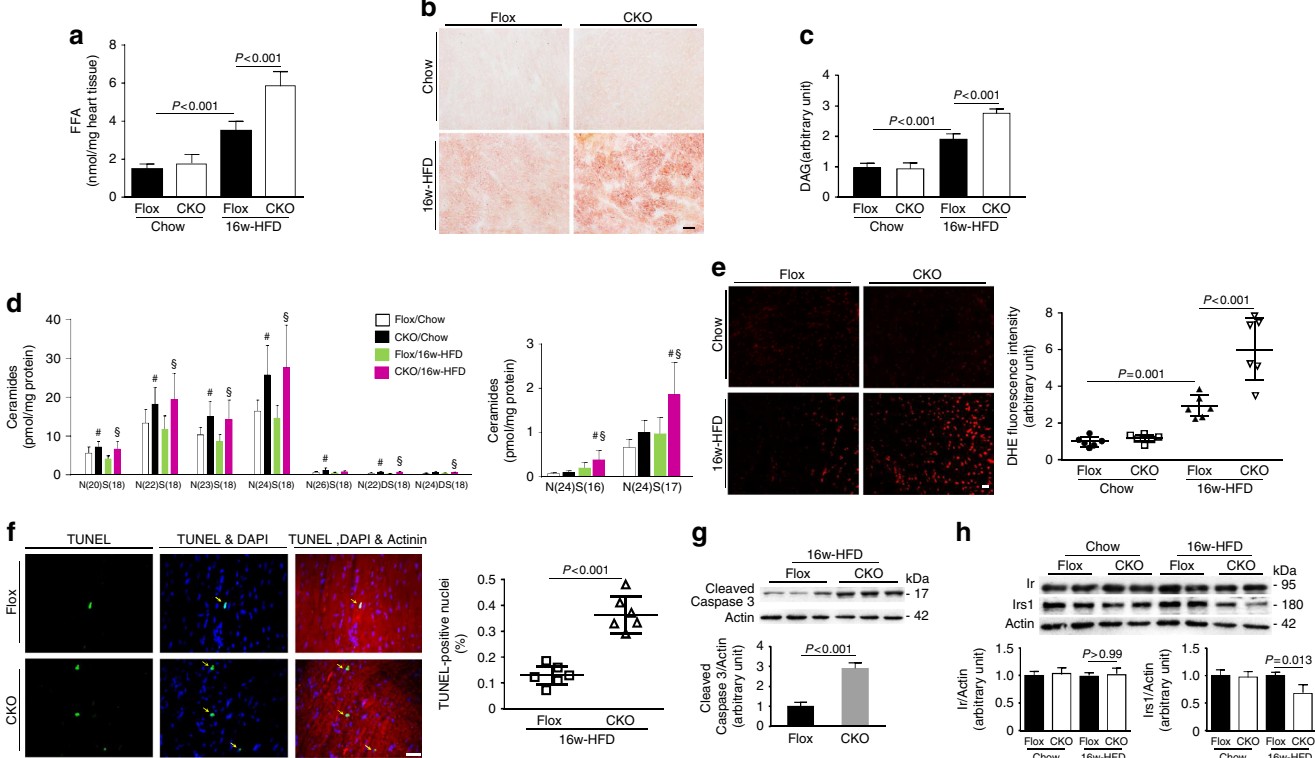

**Fig. 4** Loss of Erk5 caused cellular injury in lipids overloaded hearts. **a** FFA extracted from heart tissues, and **b** Heart sections stained with Oil Red O, indicated that neutral lipids were increased in HFD-CKO hearts (*scale bar*: 20 μm). **c** Measurement of DAG content demonstrated an accumulation of its level in HFD-CKO hearts. **d** Triple quadrupole mass spectrometer analysis of ceramide species showed an altered ceramide species profile in CKO hearts. #$P < 0.05$ vs. chow-Flox, §$P < 0.05$ vs. HFD-Flox, *$P < 0.05$ vs. chow-CKO, $n = 7$ mice per group. **e** DHE staining determined more superoxide generation in HFD-CKO hearts (*left panel, scale bar*: 20 μm). Fluorescence intensity was quantified (*right panel*). **f** TUNEL assay by triple staining with DAPI (*blue*), anti-α-actinin antibody (*red*), and TUNEL (*green*) detected apparent apoptosis in HFD-CKO hearts (*left panel, scale bar*: 20 μm), arrows indicated TUNEL positive nuclei. The quantification of TUNEL positive nuclei is shown in bar graphs (*right panel*), $n = 6$ mice per group. **g** Immunoblot analysis showed increased active caspase-3 in the hearts of HFD-CKO mice. The quantifications are represented by the bar graphs. **h** Immunoblot analysis showed decreased Irs1 in HFD-CKO hearts. The quantification is represented by the bar graphs, $n = 6$ mice. Actin is the protein loading control. Data are presented as means ± SD

with reduced number (Fig. 3b). The morphology displayed notable heterogeneity in fragmentation, shrinking, or swelling. Higher magnification showed an increased area of disarrayed cristae, the appearance of vacuoles and a reduced electron density of the matrix in the disrupted mitochondria (Fig. 3c). In addition, the lipid droplets density was higher in HFD-CKO hearts compared with HFD-Flox hearts (Fig. 3c). In contrast, no alterations in mitochondrial number and structure were discovered in chow-fed groups (Fig. 3b, c). Given these profound aberrations in mitochondrial structure, we next assessed electron transport chain (ETC) complex activity and ATP production. We measured substrate-driven oxygen consumption in muscle fibers of the working hearts. Fiber respiration rates of chow hearts were not different (Fig. 3d). Compared with HFD-Flox mice, cardiac fiber OXPHOS capacity (respiration rate at saturating levels of ADP) with complex I-linked substrates (glutamate/malate) was significantly reduced in HFD-CKO hearts (Fig. 3d), but not with complex II-linked (succinate/rotenone) or complex IV-linked (TMPD and ascorbate) substrates. This indicated a decrease in the oxidation of Nadh2 at complex I. As a result of reduced complex I function, ATP production and the respiratory control ratio (RCR, a proxy for mitochondrial efficiency of ATP production) were consistently decreased in HFD-CKO hearts by 52 and 42%, respectively (Fig. 3e). Furthermore, we assessed complex I activity in mitochondrial homogenates using fatty acid (palmitoylcarnitine/malate) as substrates. As anticipated,

Erk5-deficient mitochondria prepared from HFD-fed mice showed a reduction in FA-driven respiration (Fig. 3f) compared with their wild-type counterparts. We went on to examine oxidation capacity. Enzymatic activity of β-hydroxylacyl CoA dehydrogenase and pyruvate dehydrogenase was reduced in HFD-CKO hearts, indicating a notable suppression in both fatty acid and glucose oxidation (Fig. 3g). In contrast, HFD-Flox hearts displayed an increased FAO compared with chow-fed groups (Fig. 3g).

**Overload lipids cause cellular injury in Erk5-CKO hearts.** Mitochondrial FAO capacity is coupled with FFA uptake, and when the balance is disrupted by suppressed oxidation, it can lead to lipid deposition in the cytosol in association with a series of cellular injuries, including oxidative damage[13, 14]. We first measured myocardial FFA content, which was copiously increased by 40% in HFD-CKO hearts compared with HFD-Flox hearts (Fig. 4a). Similarly, tissue sections of HFD-CKO hearts revealed obvious lipid accumulation, evidenced by Oil Red O staining, compared with HFD-Flox hearts (Fig. 4b). An accumulation of lipid intermediates, like diacylglycerol (DAG) and ceramides, is known to increase ROS production through damage to the mitochondrial inner membrane integrity and enhanced Nadph oxidase activity[15, 16]. Therefore, we detected DAG content and found an accentuated level in HFD-CKO

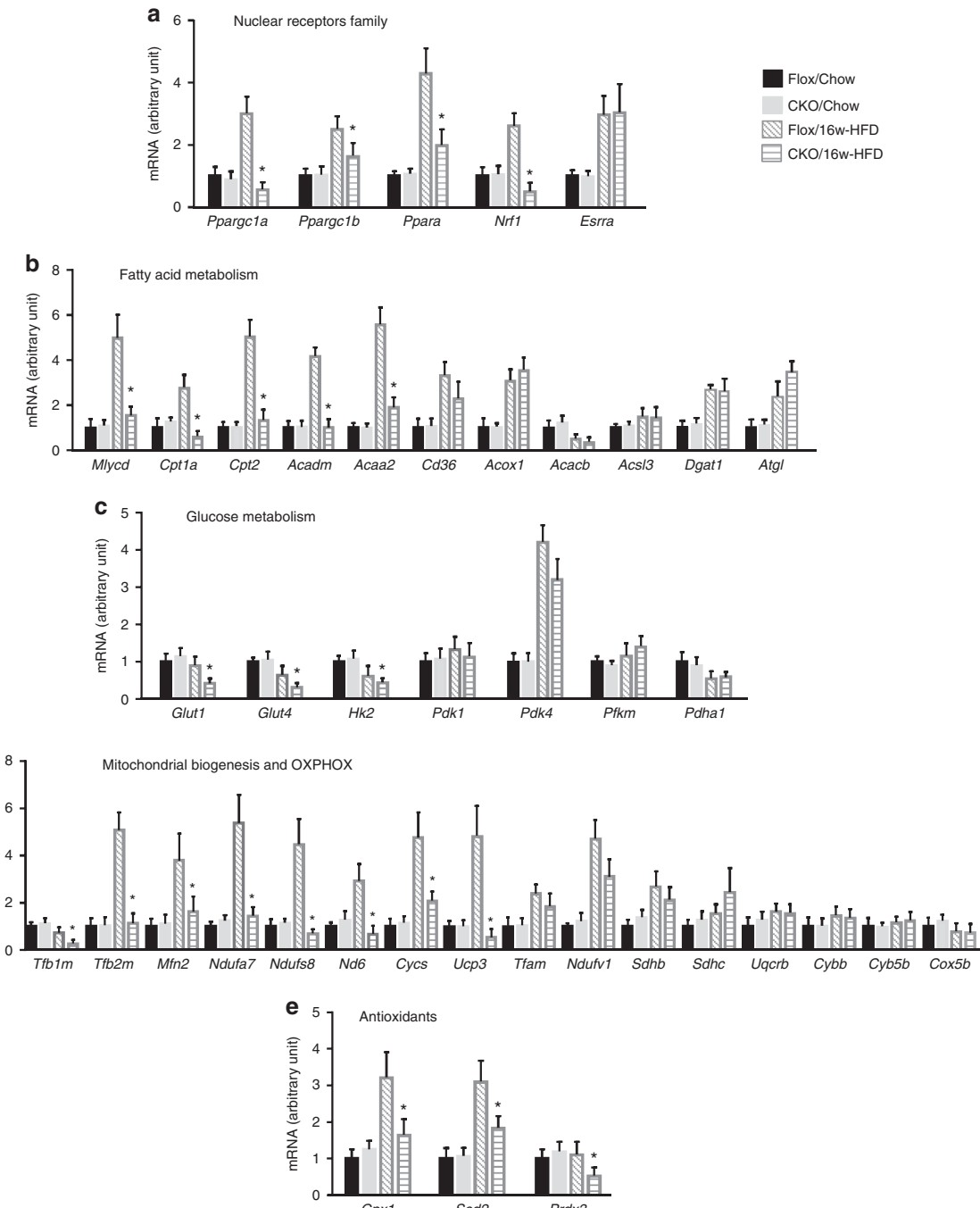

**Fig. 5** Erk5 was required for mitochondrial function-related gene expression programmes. **a** Quantitative real-time PCR (qPCR) showed a decreased expression of *Ppargc1a*, *Ppargc1b*, *Ppara*, and *Nrf1* in the hearts of HFD-CKO mice. The mRNA level of genes involved in FAO **b**, glycolytic action **c**, or mitochondrial biogenesis and OXPHOS **d** were assayed by qPCR. **e** The mRNA level of antioxidants was decreased in HFD-CKO hearts. Data are plotted as means ± SD (*$P < 0.05$, vs. HFD group, $n = 7$ mice per group)

hearts (Fig. 4c). We made a further attempt to assess ceramide profile using tandem mass spectrometer and observed a differential profile of ceramide species in CKO hearts versus Flox hearts. 9 species were detected in the hearts. Among these species, the level of 6 species was elevated in chow-CKO compared with chow-Flox hearts, the level of 8 species was higher in HFD-CKO compared with HFD-Flox hearts, and the amount of CER[N(24)S (16)] and CER[N(24)S(17)] was augmented in HFD-CKO hearts compared with HFD-Flox and chow-CKO hearts (Fig. 4d). Next, we evaluated oxidative stress in HFD-CKO hearts.

Dihydroethidium (DHE) staining for cytosolic superoxide demonstrated an increased ROS production in HFD-CKO hearts (Fig. 4e). Meanwhile, apoptosis occurrence was examined by TUNEL assay, and active caspase-3 expression also confirmed unambiguous oxidative damage in HFD-CKO hearts (Fig. 4f, g). Overload of lipid metabolites is associated with insulin resistance[13]. We finally examined the expression of insulin receptor (Ir) and insulin receptor substrate-1 (Irs1). Decreased expression of Irs1 in HFD-CKO hearts suggested a compromised insulin action (Fig. 4h). Together, the phenotypic analysis clearly

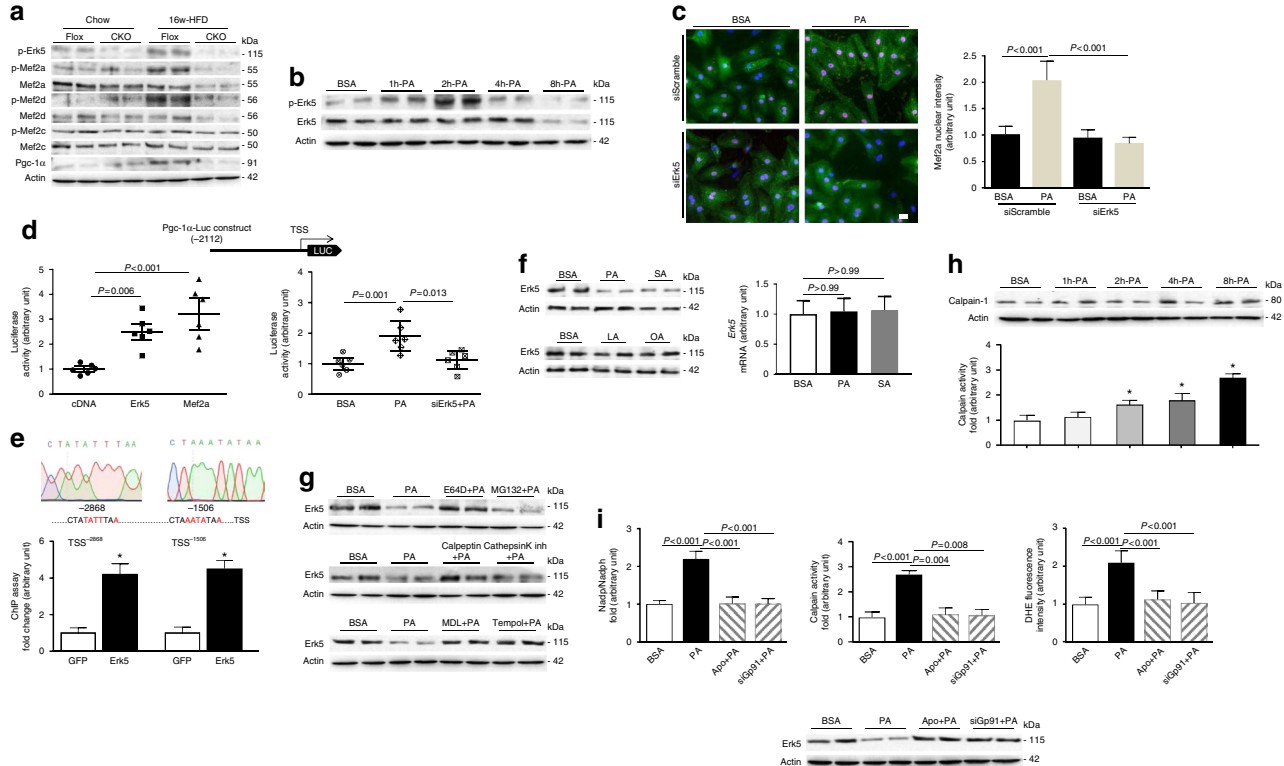

**Fig. 6** Molecular basis underlying Erk5 regulation of Pgc-1α and Erk5 degradation. **a** Immunoblot analyses revealed the phosphorylation level and total expression of Erk5, Mef2a, Mef2c, Mef2d, and Pgc-1α in the hearts of Flox and CKO mice fed with chow or HFD for 16 weeks. **b** Immunoblot analyses showed increased activation of Erk5 in NRCMs treated with palmitate acid (PA, 500 μM) for 2 h; while the total expression of Erk5 is significantly decreased by PA for 8 h. **c** NRCMs were stained with DAPI (*blue*), anti-α-actinin antibody (*green*), and Mef2a (*red*) (*scale bar*: 20 μm), fluorescence intensity of Mef2a in nuclei was quantified. **d** Increased Pgc-1α reporter luciferase activity was detected with overexpression of Erk5 or MEef2a. The increased reporter activity induced by palmitate was blunted by knockdown of Erk5 in NRCMs. **e** ChIP analysis demonstrated that enhanced binding of Mef2a to the *Ppargca* (Pgc-1α) promoter region at two sites (TSS, transcriptional starting site, -1506 and -2868, marked in *red*) in HL-1 cardiomyocytes is Erk5-dependent. Sequence analysis of the purified PCR fragment bound by anti-Mef2a antibody confirmed the Mef2 consensus binding sequence. **f** Immunoblot analyses showed decreased Erk5 in ARCMs treated with 8 h PA or stearate acid (SA, 500 μM), but not by unsaturated FFA linoleate acid (LA, 500 μM) or oleate acid (OA, 500 μM). qPCR analysis showed no reduction in mRNA expression of *Erk5* in PA or SA-treated ARCMs. **g** Immunoblot analyses showed that degradation of Erk5 was blocked by pretreatment of E-64D (25 μM, 4 h), but not by MG132 (10 μM, 4 h). Erk5 degradation was caused by calpain-1 and its expression could be restored by calpeptin (20 μM, 6 h), MDL-28170 (30 μM, 16 h) or Tempol (10 nM, 16 h), but not by cathepsin K inhibitor II (1 μM, 6 h). **h** Calpain-1 expression and its activity were measured. **i** Increased calpain activity, Nadph oxidase activity and DHE fluorescence intensity induced by PA were inhibited by pretreatment of apocynin (10 μM, 1 h) or gp91phox knockdown. Application of apocynin or Gp91phox knockdown restored Erk5 expression despite palmitate stimulation, $n = 5–6$ independent experiments per group. Actin is the protein loading control. Data are presented as means ± SD

reveals that Erk5 loss in the heart rendered the mice unable to cope with excess dietary fat-induced metabolic stress.

**Erk5 regulates Pgc-1α controlled gene expression programmes.** Given the profound impact of Erk5 deficiency on mitochondrial functions under dietary fat stress, we hypothesized that Erk5 is a critical signaling component upstream of mitochondrial function-related gene expression programmes. We first ascertained the relationship between Erk5 and mitochondrial transcriptional regulators. Expression of *Ppargc1a*, *Ppargc1b*, *Ppara*, and *Nrf1*, but not *Esrra* was decreased HFD-CKO hearts, compared with HFD-Flox hearts (Fig. 5a). We next screened a large body of key genes responsible for FAO, glycolytic action, mitochondrial biogenesis, OXPHOS, and antioxidant capacity. By qPCR analysis of FAO genes, we found that expression of *Mlycd*, *Cpt1a*, *Cpt2*, *Acadm*, and *Acaa2* was significantly down-regulated in HFD-CKO hearts compared with HFD-Flox hearts, whereas expression of *Cd36*, *Acox1*, *Acacb*, *Acsl3 Dgat1*, and *Atgl* remained comparable (Fig. 5b). In relation to glycolytic

capacity, we observed less expression of *Glut1*, *Glut4*, and *Hk2*, but not *Pdk1*, *Pdk4*, *Pfkm*, and *Pdha1* in HFD-CKO hearts (Fig. 5c). Further assessment of 16 genes for mitochondrial biogenesis and OXPHOS showed that expression of *Tfb1m*, *Tfb2m*, *Mfn2*, *Ndufa7*, *Ndufs8*, *Nd6*, *Cycs*, and *Ucp3* was reduced in the hearts of HFD-CKO mice (Fig. 5d). In addition, expression of antioxidants *Gpx1*, *Sod2*, and *Prdx3* was decreased in HFD-CKO hearts (Fig. 5e). This widespread alteration in Pgc-1α-mediated genes supports our assumption that Erk5 is involved in mitochondrial function-related gene expression programmes.

**Erk5/Mef2 cascade positive regulation of Pgc-1α.** To probe the molecular basis underlying the role of Erk5 in response to FFA stimulation, we first determined Erk5 activation by examining its phosphorylation level. In HFD-Flox hearts, Erk5 phosphorylation at Thr218/Tyr220 was increased along with elevated phosphorylation of Mef2a and Mef2d. Conversely, in HFD-CKO hearts, expression of Mef2a and Mef2d was diminished concomitantly with a diminution in their

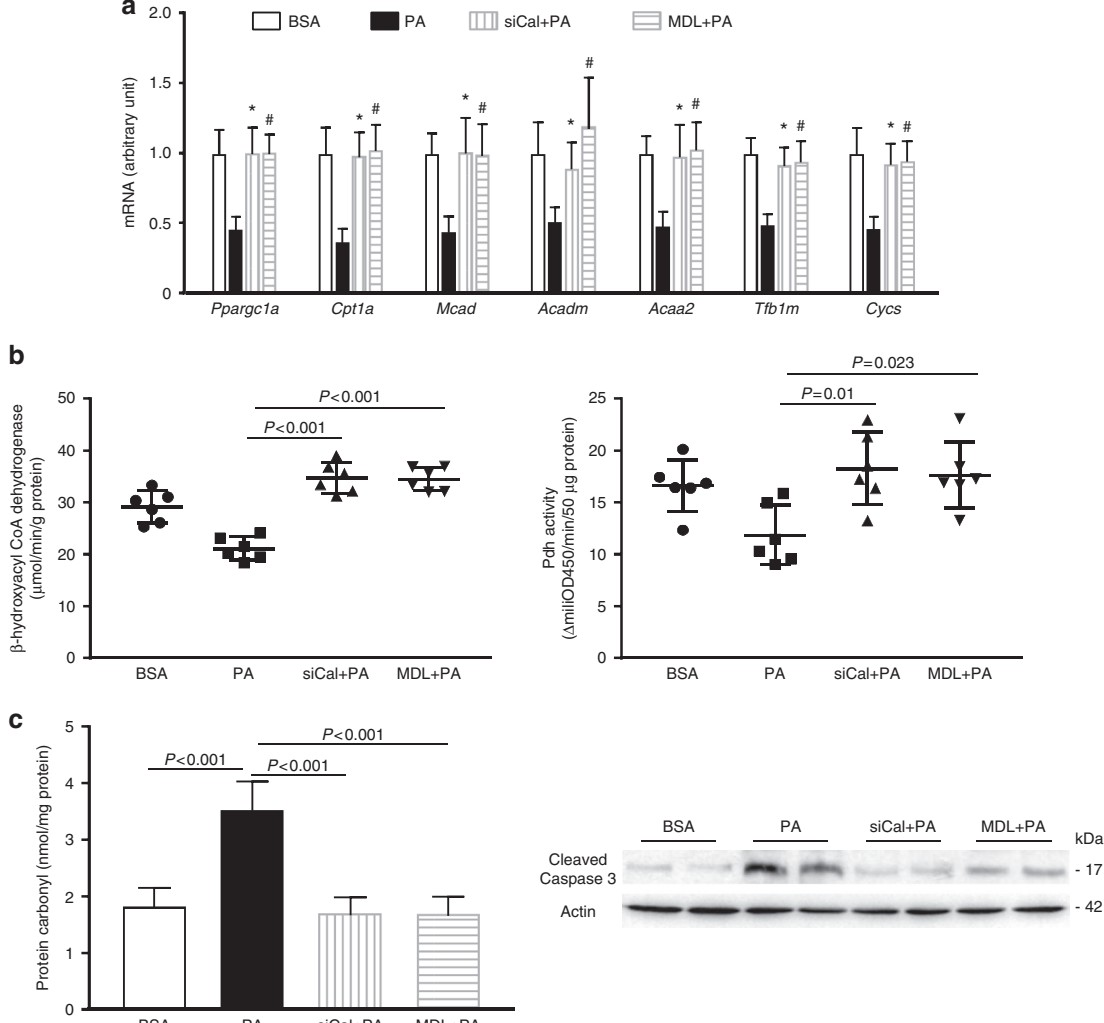

**Fig. 7** Prevention of Erk5 degradation rescued mitochondrial functions in ARCMs. **a** Prevention of Erk5 reduction by calpain-1 knockdown or MDL-28170 (30 μM for 16 h) was able to retrieve the mRNA level of key mitochondrial genes reduced by 8 h PA treatment. * or #$P < 0.05$, vs. PA-treated group. **b** Mitochondrial oxidation activities of β-hydroxylacyl CoA dehydrogenase and pyruvate dehydrogenase (Pdh) were restored when calpain-1 was blocked. **c** FFA-induced oxidative damage was ameliorated followed by calpain-1 knockdown or inhibition, indicated by decreased protein carbonyl amount or active caspase-3 expression, respectively. $n = 6$ independent experiments per group. Actin is the protein loading control. Data are presented as means ± SD

phosphorylation level; in addition, Pgc-1α expression was reduced in HFD-CKO hearts. However, the phosphorylation level and total expression of Mef2c remained unchanged (Fig. 6a). Furthermore, stimulation of neonatal rat cardiomyocytes (NRCMs) for 2 h with palmitate significantly increased Erk5 phosphorylation (Fig. 6b). Subsequent to Erk5 activation, Mef2a activity indicated by its nuclear composition was observed following palmitate, whereas knockdown of Erk5 deprived palmitate-induced Mef2a nuclear accumulation (Fig. 6c). Next, Pgc-1α luciferase reporter determined whether Erk5/Mef2 cascade is requisite for Pgc-1α activity. Pgc-1α luciferase activity was appreciably intensified in NRCMs with overexpression of Erk5 or Mef2a (Fig. 6d). Consistently, palmitate-induced Pgc-1α luciferase activity, whereas knockdown of Erk5 abolished this activity (Fig. 6d). Two Mef2 binding sites are located at the proximal promoter region (relative to transcription start site, -1506; -2868) of murine Pgc-1α[17]. Indeed, ChIP assay in HL-1 cardiomyocytes showed that Erk5 overexpression leads to increased Mef2a binding of the Pgc-1α promoter at the two binding sites (Fig. 6e). Together, these data demonstrate that the Erk5/Mef2 cascade positively regulates Pgc-1α activity under FFA stress.

**Erk5 restoration rescues mitochondrial functions**. Next, we attempted to gain mechanistic insight into how Erk5 is decreased by prolonged FFA stimulation. Adult rat cardiomyocytes (ARCMs) were treated with saturated fatty acid, palmitate or stearate, which resulted in a substantial reduction in Erk5 protein expression, but not in its mRNA expression, which is consistent with our observation in obese/diabetic hearts. However, treatment with unsaturated FFA (oleate and linoleate) had no effect on Erk5 expression (Fig. 6f). Meanwhile, we noticed a reduction in both mRNA and protein expression of Mef2a and Mef2d by palmitate (Supplementary Fig. 7). Saturated FFA can induce protein degradation by proteasome-mediated pathways or by proteolysis often through cysteine proteases; therefore, we treated ARCMs with MG132 (proteasome inhibitor) or E-64D (cysteine protease inhibitor). Interestingly, E-64D, but not MG132, prevented palmitate-induced Erk5 protein degradation (Fig. 6g). Calpain and cathepsin are E-64D-sensitive cysteine proteases in the heart. We discovered that inhibition of calpain-1 (major cardiac isoform) by calpeptin, not cathepsin K by cathepsin K inhibitor II, restored Erk5 expression in cycloheximide pretreated ARCMs (Fig. 6g). We further determined that the calpain-1 pharmacological inhibitor (MDL-28170) or ROS scavenger

(Tempol) to restored Erk5 expression despite palmitate (Fig. 6g), which provides substantial evidence of Erk5 degradation is caused by calpain-1, whose activation is triggered by ROS. Although the expression of calpain-1 was not affected by palmitate, its activity was increased gradually (Fig. 6h). Using a cell-free system, we

observed recombinant calpain-1 was able to rapidly degrade recombinant Erk5 (Supplementary Fig. 8a). Furthermore, we used a cell-based system with transfection of Flag-tagged human Erk5 in COS-7 cells. Flag-Erk5 was purified by immunoprecipitation prior to the incubation with human recombinant calpain-1.

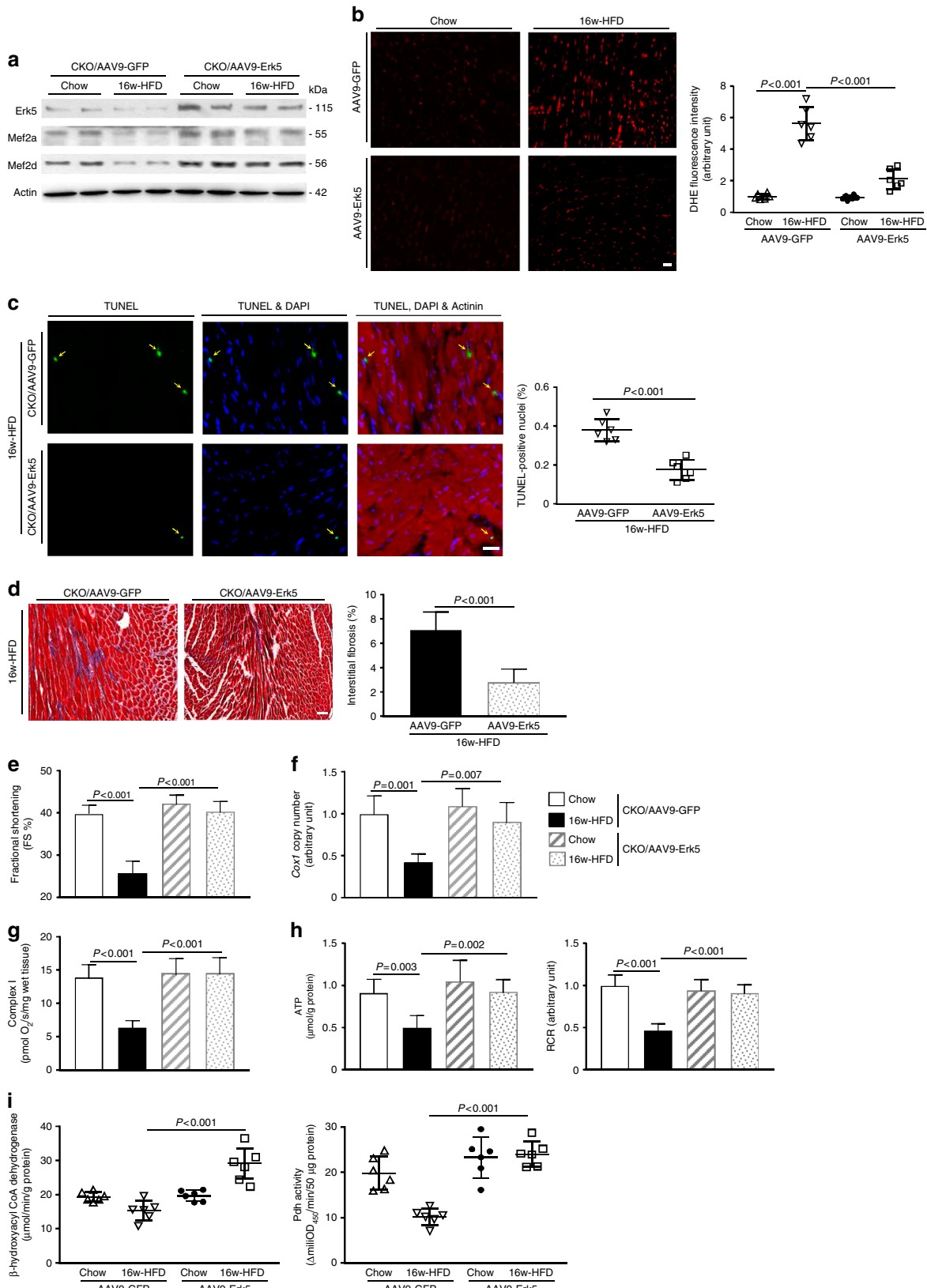

Multiple bands ranging from 55 to 25 KD were captured by immunoblotting, indicating multiple calpain-1 cleavage sites are present in the Erk5 sequence (Supplementary Fig. 8b). These observations provide direct evidence of Erk5 degradation by calpain-1. We next tested whether ROS-triggered calpain-1 activation is dependent on Nadph oxidase activity. Nadph oxidase inhibitor, apocynin, inhibited palmitate-induced calpain activity and Nadph oxidative activity (Fig. 6i). Similar effects were observed when Gp91phox (major Nadph oxidase in cardiomyocytes) was knocked down in ARCMs (Fig. 6i). As a result, application of apocynin or Gp91phox knockdown restored Erk5 expression also alleviated ROS production (Fig. 6i). Interestingly, mitochondria-targeted antioxidant (coenzyme Q10) or xanthine oxidase inhibitor (Allopurinol) did not exert an effect on calpain activity (Supplementary Fig. 9), indicating that Nadph oxidase is likely the primary source of oxidative stress induced by FFA. In addition, we observed increased calpain activity and Nadph oxidase activity in CKO hearts of 16-week HFD, but protein expression of calpain-1 and Gp91phox remained unchanged (Supplementary Fig. 10). Furthermore, to investigate whether prevention of Erk5 degradation would rescue mitochondrial function under FFA stress, we knocked down calpain-1 or applied MDL-28170 in ARCMs. Convincingly, we observed normal levels of key mitochondrial genes, such as *Ppargc1a*, *Cpt1a*, *Mcad*, *Acadm*, *Acaa2*, *Tfb1m*, and *Cycs*, despite palmitate treatment (Fig. 7a). Functional analysis of fuel oxidation measured using acetoacetyl CoA and pyruvate as substrates revealed a normal metabolic state (Fig. 7b). As oxidative damage indicators, total protein carbonylation and active caspase-3 expression were significantly reduced along with Erk5 restoration (Fig. 7c). Furthermore, we evaluated whether Erk5 positive regulation of mitochondrial function is through Pgc-1α. As anticipated, palmitate caused a downregulation of key mitochondrial genes and decreased mitochondrial fuel oxidation in Erk5-deficient ARCMs, whereas Pgc-1α overexpression alleviated such aberrations (Supplementary Fig. 11). Finally, we sought evidence whether restoration of Erk5 in CKO hearts would rescue metabolic stress-induced cardiomyopathy. The cardiac expression of Erk5 was achieved by the AAV9-delivery of a troponin T (TnT) promoter-driven Erk5 (AAV9-Erk5, $1 \times 10^{11}$ genomic particles for injection) (Supplementary Fig. 12). Thereafter, AAV9-GFP (as the control) or AAV9-Erk5 injected CKO mice were subject to 16-week chow diet or HFD. Re-establishment of Erk5 expression and up-regulation of Mef2a and Mef2d in the myocardium receiving AAV9-Erk5 injection were observed (Fig. 8a). Cardiac morphology, cardiac function, mitochondrial functions and expression of key mitochondrial genes were evaluated. As anticipated, Erk5 restoration in CKO hearts alleviated the pathological responses to HFD as evidenced by reduced ROS level, apoptosis occurrence and fibrosis content (Fig. 8b–d). Consequently, cardiac contractility was improved (Fig. 8e). Meanwhile, mtDNA was restored in the AAV9-Erk5 injected HFD-CKO hearts (Fig. 8f). Consistently, mitochondrial complex I

respiration was increased attendant with a rise in ATP production, RCR and mitochondrial oxidation capacity (Fig. 8g–i) in these mice. In addition, the transcript levels of *Ppargc1a*, *Cpt1a*, *Acadm*, *Acaa2*, *Tfb2m*, *Nd6*, *Cycs*, and *Sod2* were upregulated in the AAV9-Erk5 injected myocardium (Supplementary Fig. 13). Taken together, our data show that prevention of Erk5 degradation or Erk5 restoration is able to rescue mitochondrial function and alleviate oxidative injury induced by prolonged FFA stress.

## Discussion

The key findings of this study are: (1) Erk5/Mef2 signaling upstream of Pgc-1α is responsible for positive regulation of mitochondrial function-related gene expression programmes; (2) Cardiac Erk5 loss in the HFD-fed mice causes contractile dysfunction and impaired mitochondrial function; (3) Gp91phox activation of calpain-1 degrades cardiac Erk5 in the face of prolonged FFA stress. Restoration of Erk5 expression by the AAV9 approach alleviates mitochondrial dysfunction and subsequent maladies, revealing a potential new strategy to mitigate metabolic cardiomyopathy from obesity and diabetes (Fig. 9).Transcription factors responsible for the control of Pgc-1α gene expression include Mef2, Creb, and Klf4[17–19]. Mef2 is able to activate Pgc-1α through a positive feedback loop when Pgc-1α works as a co-activator for itself activation[20]. Interestingly, total deletion of Mef2a in mice resulted in a perinatal death, which was ascribable to significant deprivation of mitochondria[21]. These abnormalities bare some phenotypic similarities to those in mice with double deletion of Pgc-1α and Pgc-1β[22]. Camk activation of Pgc-1α expression involves Creb[18]. Moreover, the transducer of regulated Creb-binding protein (Torc) may enhance Creb-dependent Pgc-1α transcription[23]. A recent study by Liao et al. reported that Klf4 is required for optimal function of the Pgc-1/Err transcriptional complex[19]. In addition, p38 and Ampk can modulate Pgc-1α activity through post-translational mechanisms[24–26]. Of note, expression of the aforementioned signaling molecules remains unchanged in the obese hearts, although phosphorylation of p38 and Ampk altered in the present study. It is interesting to point out that the molecular mechanisms underlying the loss of Pgc-1α in heart failure are poorly understood. This study is focused on Erk5, and our data with Erk5 loss may fill in this knowledge gap of Pgc-1α falling by presenting a previously unappreciated mechanism in the FFA-stressed heart.

Among Erk5 downstream effectors, Mef2 family of transcriptional factors is well defined with Mef2a and Mef2d being major cardiac isoforms[21, 27]. In concert, we found that alongside Erk5 loss, reduced expressions of Mef2a and Mef2d at transcriptional and protein levels were apparent, but not Mef2c, Creb, and Klf4. As Mef2 can be self-regulated upon its activation[28, 29], we hypothesized that reduction in Mef2a and Mef2d transcription is likely due to a lack of Erk5 activation. In addition, we did not observe changes in Mef2c that may be explained by its regulation not only through Erk5, but also via p38. In our defined cellular

---

**Fig. 8** Erk5 protects against metabolic stress-induced cardiomyopathy. **a** The expression of Erk5, Mef2a, and Mef2d were examined in the hearts from CKO mice injected with AAV9-GFP or AAV9-Erk5 followed by 16-week chow or HFD feeding. Actin is the protein loading control. **b** DHE staining indicated less ROS production in AAV9-Erk5-HFD hearts (*left panel, scale bar*: 20 μm). Fluorescence intensity was quantified (*right panel*). **c** TUNEL assay by triple staining with DAPI (*blue*), anti-α-actinin antibody (*red*), and TUNEL (*green*) determined fewer apoptosis in AAV9-Erk5 injected HFD-CKO hearts (*left panel, scale bar*: 20 μm), arrows indicated TUNEL positive nuclei. Quantification of TUNEL positive nuclei is represented by the bar graphs (*right panel*). **d** Masson's trichrome staining detected less interstitial fibrosis in AAV9-Erk5 injected HFD-CKO hearts (*left image, scale bar*: 50 μm). Quantification of the fibrosis area is shown in the bar graphs (*right panel*). Restoration of Erk5 in CKO hearts improved **e** cardiac function and mitochondrial function evidenced by **f** increased the content of mtDNA, **g** enhanced complex I respiration rate, **h** elevated ATP production along with RCR, and **i** improved mitochondrial oxidation capability indicated by β-hydroxylacyl CoA dehydrogenase and pyruvate dehydrogenase (Pdh). $n = 6$ mice per group. Data are presented as means ± SD

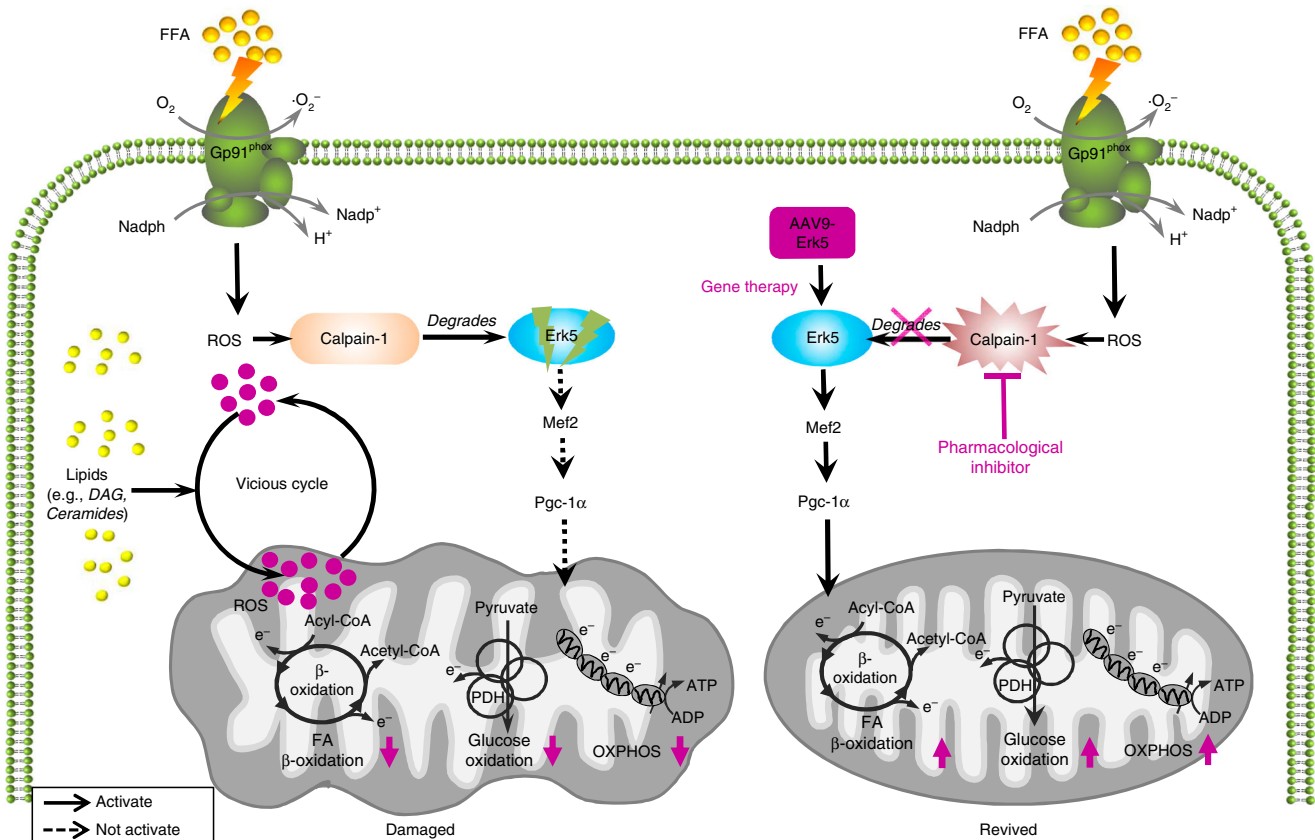

**Fig. 9** Schematic model of Erk5 protection against metabolic stress. FFA induces escalated ROS production from Gp91phox, which activates caplain-1, leading to a breakdown of Erk5. Loss of Erk5 and resultant Pgc-1α downregulation impose detrimental impacts on mitochondrial functions through accumulation of toxic lipids and vicious cycle of ROS. Prevention of Erk5 degradation or Erk5 restoration by AAV9 approach revives mitochondrial functions

system, overexpression of either Erk5 or Mef2a led to increased Pgc-1α luciferase reporter activity. Furthermore, Erk5 overexpression led to enhanced endogenous Mef2a binding of the Pgc-1α promoter. These molecular results establish a signaling cascade by Erk5/Mef2 for positive regulation of Pgc-1α. It is noteworthy that cardiac mitochondrial function in Erk5-CKO mice fed by a chow diet was preserved with a normal level of cardiac Pgc-1α, suggesting Erk5 is a prerequisite for regulating Pgc-1α when faced with a stress challenge. This notion has been corroborated by the findings that Pgc-1α overexpression was able to alleviate adverse effects induced by Erk5 knockdown in combination with FFA stimulation in cardiomyocytes. Interestingly, analogous to Erk5-CKO mice being succumbed to pressure overload, transverse aortic constriction led to profound cardiac failure in Pgc-1α$^{-/-}$ mice, whilst non-stressed Pgc-1α$^{-/-}$ mice displayed normal cardiac function despite reduced cardiac FAO and OXPHOS genes[30]. These findings imply that mitochondrial capacity is sufficient for maintaining myocardial energy and contractile function when confronted with reduced expression or activity of these transcriptional regulators. However, in the presence of superimposed stress, mitochondrial activity reaches a critical threshold where it can no longer cope with increased workload, eventually leading to mitochondrial dysfunction and impaired cardiac function. This theory may explain why Erk5-CKO and Pgc-1α$^{-/-}$ mice behave normally under non-stressed conditions, but fail to handle stress challenges.

It is generally believed that hyperlipidemia occurring with obesity may increase cardiac lipid uptake and formation of ceramides, DAG and other lipids, which initially activates membrane-bound Nadph oxidases (NOXs) to generate ROS[13, 15, 16]. Consequently, the initial increase in cytosolic ROS enhances the activity of Pgc-1α/Pparα as an adaptive reaction, which shifts the balance of energy metabolism toward greater fatty acid oxidation to generate more ATP. At this point, increased delivery of reducing equivalents from FAO is not always synchronized with up-regulation of OXPHOS gene expression, therefore causing mitochondrial uncoupling that instigates de novo ROS generation within the mitochondria. If metabolic stress continues, Pgc-1α levels eventually fall, which leads to dampened fatty acids and glucose oxidation. Therefore, a large amount of offensive lipids accumulate in the cytosol, which further intensifies NOXs-generated ROS production. The ROS from NOXs amplifies mitochondrial ROS production through a mechanism called "ROS induced ROS release"[31]. Amplification of ROS through this vicious cycle damages mitochondria, and thereafter, cardiomyocytes perish and heart failure ensues. Using animal models with excess dietary fat, our study demonstrates that Erk5 is likely an intermediate signal linking FFA-stimulated ROS signaling to the regulation of Pgc-1α, and we explain how prolonged FFA-caused Erk5 loss contributes to the development of cardiomyopathy. When the heart is in response to acute metabolic stress, up-regulation of Pgc-1α/Pparα by the initial cytosolic resource of ROS requires the Erk5/Mef2 cascade. However, when FFA stress becomes unrelenting, Gp91phox activates caplain-1, leading to a breakdown of Erk5. Loss of cardiac Erk5 causes Pgc-1α downregulation, which imposes detrimental impacts on mitochondrial function.

Therefore, ill-fated mitochondria struggle to produce sufficient energy, a vicious cycle of ROS production is instigated, cell death ensues, and cardiomyopathy progressing to failing stage becomes inevitable. Given such a convoluted pathological process, our cardiac Erk5 deletion model was used to investigate this disease condition. Furthermore, we restored Erk5 expression by an AAV9 mediated overexpression in CKO mice that prevented the development of cardiomyopathy. It is necessary to point out that mouse models with knockout or overexpression of a protein are not perfect experimental models. The most common drawback is widespread gene remodeling as compensatory responses, which could cause confounding effects and hamper precise interpretations of the protein functions. Despite this pitfall, mouse models are still invaluable for revealing underlying biological mechanisms through fine-tuned models and multidisciplinary technologies. In the case of Erk5, cardiac deletion of Erk5 does not manifest heart developmental defects[10], but rather, knockouts display diet-induced cardiac pathology reminiscent of metabolic cardiomyopathy. Moreover, virus-based restoration of Erk5 avoids potential compensatory effects caused by the genetic modification of Erk5 overexpression. The combination of the two mouse models and our cellular study has revealed the critical role of Erk5 in the FFA-stressed heart.

On a pertinent note, altered ceramide species profile and increased DAG level have been detected in HFD-CKO hearts. It is needed to clarify that although we observed these changes, however we cannot ascertain that these lipids are toxic responsible for diet-induced cardiac pathology. For example, DAG is an important intracellular secondary messenger, which triggers signal transduction cascades to regulate a wide range of biological functions through Pkc activation or other downstream effectors. Likewise, ceramide species are also involved in modulation of various intracellular events. Therefore, future studies will be required to assess the role of individual enzymes for synthesis or hydrolysis of these lipids in the CKO hearts that could provide tangible evidence on whether DAG and certain ceramide species play a role in pathogenesis of metabolic cardiomyopathy. It is interesting to point out that we detected reduced complex I in both myocardial fibers and mitochondrial homogenates of HFD-CKO hearts. Complex I is the largest complex with multi-step assembly process and the rate-limiting step of the mitochondrial OXPHOS system. Due to its complexity in assembly and proximity to mitochondria-generated ROS, complex I stands vulnerable to damage, and its defective activity is a critical determinant of impaired OXPHOS capacity. In our study, the loss of Erk5 caused a significant downregulation of Pgc-1α-dependent transcription programmes. As a result, a wide array of mitochondrial function genes was decreased, including important complex I subunits, *Ndufa7*, *Ndufs8*, *Nd6*, and mitochondrial biogenesis genes, such as *Tfb1m*, *Tfb2m*, and *Mfn2*. Those reductions gave rise to impaired complex I activity and decreased mitochondrial numbers, respectively.

In our previous study, we reported that the mice defective of cardiac Erk5 failed to resist pressure overload, manifesting cardiac dysfunction and less hypertrophy, succumbing to heart failure due to cumulative cardiomyocyte death[10]. Consistently, we also observed less hypertrophy induced by HFD in CKO mice in the current study. A recent study by Appari et al. provided further evidence linking Erk5 to hypertrophy[32]. In the C1q-tumor necrosis factor-related protein 9 (Ctrp9) knockout mouse model, they showed a reduced level of Erk5 after TAC, while Ctrp9 overexpression entailed increased Erk5 activation. Collectively, these findings duly demonstrate the requirement of Erk5 in development of hypertrophy in response to various pathological stresses. However, future study will be needed to

investigate Erk5 overexpression mouse model that can directly establish pro-hypertrophic status of Erk5 in the heart.

Insulin resistance by impaired Pkb signaling is a key feature of metabolic stress-induced cardiomyopathy[13]. HFD-CKO hearts manifested suppressed glucose oxidation and decreased expression of multiple components of glycolytic capacity, such as; *Glut1*, *Glut4*, and *Hk2*, as well as decreased Irs1 expression. However, it is currently unknown whether this impairment is secondary to suppressed FAO or Erk5 holds a tangible role in insulin signaling. A previous study reported that sustained activation of Foxo1 or Foxo3 leads to diminished insulin responsiveness and impaired glucose metabolism in cardiac myocytes, and Foxo1 activation triggers diabetic cardiomyopathy through downregulation of Irs1 in mice[33]. Interestingly, an earlier study illustrated that Erk5 deficiency was associated with reduced Pkb phosphorylation and increased Foxo3a activity in fibroblasts[34]. This potential cross-talk between Erk5 and Pkb/Foxo signaling requests future investigation in HFD-CKO hearts with regards to glucose uptake and insulin responsiveness.

The calpain family of proteases are calcium-dependent, non-lysosomal cysteine proteases. At the last count, more than 20 substrates have been identified as calpain's targets with Erk5 as a new victim adding to its ever-growing repertoire[35]. Calpain is well known for its role in regulating cardiac hypertrophy through its activation of Nfκb by degrading Ikb[35]. Transgenic mice with constitutively expressed calpastatin (endogenous inhibitor of calpain) manifested resilience against cardiac hypertrophy in response to AngII infusion, which highlighted the functional importance of calpain in the heart[36]. Previous studies reported that calpain could be activated by Nadph oxidases to induce cardiomyocyte apoptosis, which is in line with our discovery in this study[37]. The fact that Erk5 degradation is prevented by apocynin or Gp91phox knockdown serves as good evidence.

The calpain family of proteases are druggable targets. The calpain inhibitor, calpeptin, has been shown to exert beneficial effects on the control of hypertrophy in a feline model[38]. Similarly, another calpain inhibitor MDL-28170, which was also used in this study to block Erk5 degradation, exerted a remarkable degree of protection from ventricular dysfunction in a swine model[39]. Our current study provides "proof of concept" evidence that knockdown or inhibition of calpain-1 has promising protective effects by ameliorating mitochondrial dysfunction and oxidative damage resulting from prolonged FFA stress-induced Erk5 loss. Future work will be considered to test this therapeutic option in animal models. For example, whether overexpression of calpastatin, or application of calpeptin or MDL-28170, or apocynin would allow animals to withstand HFD-induced cardiomyopathy needs to be carefully evaluated.

In summary, the novel findings discovered from this study allow us to propose a new protective mechanism for cardiac mitochondrial functions by Erk5 positive regulation of Pgc-1α. Furthermore, we show that breakdown of cardiac Erk5 by Gp91phox-dependent calpain-1 activity underlies the pathological mechanism of metabolic stress-induced cardiomyopathy. Beneficial effects of preventing Erk5 breakdown or Erk5 restoration by AAV9 approach lend solid credence to the possibility of developing a new therapeutic approach for treating metabolic cardiomyopathy and associated heat failure.

## Methods
**Animal models.** Animal studies were performed in accordance with the United Kingdom Animals (Scientific Procedures) Act 1986 and were approved by the University of Manchester Ethics Committee. All laboratory mice and rats in the current study were maintained in a pathogen-free facility at the University of Manchester. As described in our previous study[10], generation of cardiomyocyte-

specific Erk5 knockout mice (referred to as Erk5-CKO) was achieved by mating Erk5-Flox mice (C57BL/6J background) with the mice expressing Cre under α myosin heavy chain (αMHC) promoter (C57BL/6J backgound). Erk5-Flox mice were used as the control for Erk5-CKO mice. For 25w feeding study, 6-week-old male C57BL/6J mice (C57BL/6JOlaHsd) were purchased from Envigo (UK). Tissue samples of db/db, ob/ob mice were provided by Dr. Kimberly Mace (University of Manchester, Harlan Laboratories) and Dr. Lee Roberts (University of Cambridge, Jackson Lab), respectively. Left ventricular tissues from rhesus monkeys with spontaneous metabolic syndrome were provided by Dr. Rui-Ping Xiao (Peking University, China)[12].

**HFD feeding**. Since 6 weeks old, male Erk5-Flox and Erk5-CKO mice were maintained on a high fat diet (HFD, 60% calories from fat, SDS, 824054) for 16 weeks, and C57BL/6J mice were on HFD for 25 weeks. Body weight and blood pressure were recorded at the interval of 4 weeks. The CODA tail-cuff system provides measurements of systolic and diastolic blood pressure, as well as mean blood pressure. Mice were measured for food intake (g/day) and nesting time (% of 24 h) for 3 days.

**Metabolic measurements**. At 4-week intervals, mice were fasted for 6 h, and tail blood was acquired for metabolic measurements. Glucose was measured with a blood glucometer (Accu-Chek Aviva, Roche). Total blood was centrifuged for 15 min at 6000×g to collect serum for cholesterol and insulin assays. According to the manufacturers' instruction, concentrations of insulin and cholesterol were measured by insulin rodent chemiluminescence ELISA kit (ALPCO, 80-INSMR-CH01) and cholesterol quantification kit (Sigma, MAK043), respectively. Measurements were obtained by a microplate reader (Gen5, Biotek).

**Construction of adeno-associated virus-Erk5**. The pSSV9-TnT-eGFP was modified by replacing GFP with Flag-tagged human Erk5 cDNA to construct the pSSV9-TnT-Erk5 plasmid. To obtain the recombinant AAV9-Erk5 and the control vector AAV9-eGFP, low passage HEK293T cells were transfected with an adenoviral helper plasmid (pDGΔVP), pAAV2-9 Rep-Cap plasmid (p5E18-VD2/9) and the adeno-associated virus (AAV) genome plasmids pSSV9-TnT-Erk5 and pSSV9-TnT-eGFP, respectively. Afterward, recombinant AAV9-Erk5 was purified by discontinuous iodixanol gradient and titrated by quantitative polymerase chain reaction (qPCR)[40]. To restore Erk5 expression in the CKO mice, the mice were anesthetized with 2% isofluorane and $1 \times 10^{11}$ genomic particles were administered by intravenous injection. AAV9-GFP was injected as the control. On the third day after the injection, the mice were fed with chow diet or HFD for 16 weeks.

**Echocardiography**. At 4-week intervals, all the experimental mice were anesthetized with 2,2,2-Tribromethanol (200 mg/kg, Sigma) and cardiac function was evaluated by echocardiography using an Acuson Sequoia C256 (Siemens) ultrasound machine. For each mouse, measurements of end-dPW, LVESd, and LVEDd dimensions were obtained as described previously[41].

**Histology**. Freshly dissected heart tissue was embedded in OCT embedding matrix (Thermo Fisher Scientific). Sections were cut at 5 or 10 μm thickness and stored in −80 °C freezer for various histological analyses. Wheat germ agglutinin staining and Masson's trichrome staining were performed on 10 μm thick cryosections[42].

**Oil Red O staining**. Cryosections of 10 μm thickness were incubated in Oil Red O working solution (0.94 g dissolved in 150 ml isopropyl alcohol and diluted in 100 ml distilled water) for 10 min at room temperature. The sections were rinsed with water for 30 min. Bright-field images were captured with a light microscope.

**Dihydroethidium staining**. In situ superoxide production was measured with the oxidative fluorescent dye dihydroethidium (DHE). Briefly, cryosections (10 μm) were incubated with DHE (10 μm) for 30 min at 37 °C. Images were captured using an Olympus fluorescence microscope. Fluorescence intensities were quantified with ImageJ software.

**TUNEL assay**. TUNEL assay was performed on cryosections using the in situ Cell Death Detection Kit (Roche). Triple staining with DAPI, anti-α-actinin antibody (Sigma 1:100), and TUNEL was performed to confirm apoptotic nuclei. An average of 10,000 myocyte cells from random fields was analyzed per animal.

**Immunohistochemistry**. Sixteen micrometer thick cryosections of hearts were fixed with 4% paraformaldehyde, followed by blocking and permeabilization with 10% normal goat serum and 0.1% Triton-X solution. The primary anti-Flag antibody (Cell Signaling, 8146 1:200) and the secondary anti-mouse antibody (Invitrogen, 1:500) conjugated to Alexa Fluoro568 (Invitrogen) were used to detect the expression of Erk5, whereas DAPI was for nuclear visualization. Images were captured using an Olympus fluorescence microscope.

**Mitochondrial respiration assessments**. Hearts were cut into longitudinal strips along fiber orientation with a diameter of 1–1.5 mm. Fiber bundles were transferred to a tube containing 2 ml ice-cold relaxation and preservation solution (2.77 mM CaK$_2$EGTA, 7.23 mM K$_2$EGTA, 20 mM imidazole, 0.5 mM dithiothreitol, 20 mM taurine, 50 mM K-MES, 6.56 mM MgCl$_2$, 5.7 mM ATP, 14.3 mM phosphocreatine, pH 7.1) with 50 μg/ml saponin (Sigma, 47036) for 20 min permeabilization. Fibers were then washed three times with fresh mitochondrial respirometry solution MiR05 (0.5 mM EGTA, 3 mM MgCl$_2$·6H$_2$O, 20 mM taurine, 10 mM KH$_2$PO$_4$, 20 mM HEPES, 1 g/l fatty acid free BSA, 60 mM potassium-lactobionate, 110 mM sucrose, pH 7.1). Respiration was measured in a two-chamber titration-injection respirometer (Oroboros, Oxygraph 2k); DatLab (Oroboros) was used for data acquisition and analysis. 2–3 mg cardiac fibers were placed in the oxygraph chambers containing MiR05 followed by addition of 10 mM glutamate and 5 mM malate for determining complex I-supported Leak respiration rate. ADP (2 mM) was added to measure the maximal phosphorylating respiration rate. After adding 0.5 mM rotenone to inhibit complex I, 10 mM succinate was used to induce complex II-supported OXPHOS capacity. Likewise, after adding 5 mM antimycin A to inhibit complex III, complex IV respiration rate was activated by 0.5 mM TMPD and 2 mM ascorbate. Finally, 10 mM cytochrome C was used to test the intactness of the outer mitochondrial membrane. To assess the efficiency of ATP production, respiratory control ratio (RCR) was calculated as OXPHOS capacity/Leak respiration rate in the presence of complex I-linked substrates (glutamate/malate).

To further determine fatty acid driven respiration in mitochondria, fresh heart tissue (5 mg) was homogenized in respiration media RM (0.5 mM EGTA, 3 mM MgCl$_2$, 60 mM K-MES, 20 mM taurine, 10 mM KH$_2$PO$_4$, 20 mM HEPES, 110 mM sucrose, and 1 g/l fatty acid free BSA, pH 7.1). 30 μl of mesh filtered sample was transferred with a Hamilton syringe to Oroboros containing RM. A total of 2 mM malate and 40 μM palmitoylcarnitine was added to record adenylate free leak respiration. A total of 5 mM ADP was subsequently added to determine maximal electron flow through fatty acid oxidative capacity. A total of 1.5 μM rotenone inhibited complex I and 2.5 μM antimycinA terminated respiration. The oxygen consumption was normalized to protein content determined by Bradford assay[43, 44].

**ATP assay**. ATP Assay Kit (Abcam, ab83355) utilizes the phosphorylation of glycerol to generate a product quantified by colorimetric (λ max = 570 nm) method. Ten milligrams of tissue was homogenized in 100 μl of ATP assay buffer and centrifuged. The supernatant was transferred for deproteinization using ice-cold 4 M perchloric acid and 2 M KOH. The assay was carried out following the manufacturer's instruction.

**Free fatty acid assay**. Free fatty acid quantification kit was purchased from Abcam (ab65341). Ten milligrams of heart tissue was homogenized with 200 μl of chloroform including 1% Triton X-100. After centrifuging, the organic phase was collected, and the chloroform was evaporated. The dried lipids were dissolved in 200 μl of Fatty Acid Assay Buffer. Free fatty acid was measured according to the manufacturer's instruction.

**DAG assay**. DAG assay kit was purchased from Cell Biolabs (MET-5028). Ten milligrams of heart tissue was homogenized in 1.5 ml methanol, followed by mixing with 2.25 ml 1 M NaCl and 2.5 ml chloroform. The mixture was centrifuged; the organic phase was dried with nitrogen. DAG was then detected following the manufacturer's instruction.

**Ceramide**. To assess the ceramide species profiling, mass spectrometry-based lipidomics assay was performed as described before[45]. Briefly, heart tissues were homogenized in ice-cold chloroform-methanol (2:1). Internal standard (50 pmol CER[N(25)S(18)] (Avanti Polar Lipids, Alabaster)) was added to each sample. The resulting lipid extract was analyzed by ultraperformance liquid chromatography (Acquity UPLC, Waters) coupled to a triple quadrupole mass spectrometer with electrospray ionization (Xevo TQ-S, Waters). Ceramides were separated using an Acquity UPLC BEH C8 column (1.7 μm, 2.1 × 100 mm, Waters) and quantified by multiple reaction monitoring assays. Data are expressed as relative amount (pmol) of ceramide per mg of tissue protein (Protein Assay II, Bio-Rad).

**Protein carbonyl ELISA assay**. The protein carbonyl elisa kit was purchased from Cell Biolabs (STA-310). BSA standards or protein lysates (10 μg/ml) extracted from ARCMs were adsorbed into a 96-well plate for 2 h at 37 °C. The protein carbonyls present in the sample or standard are derivatized to dinitrophenyl (DNP) hydrazone and detected with anti-DNP antibody (1:1000).

**Calpain activity assay and Nadph oxidase activity assay**. The fluorometric calpain activity assay (ab65308) and the colorimetric Nadp/Nadph assay (ab65349) were purchased from Abcam. Protein lysates were extracted following the manufacturer's instruction. Calpain activity was quantified by the measurement of fluorescence reading at λ max=505 nm; while the amount of Nadp and Nadph was quantified by colorimetric (OD = 450 nm) methods.

**Transmission electron microscopy**. The heart samples were fixed overnight in 0.1 M HEPES buffer (pH 7.2) containing 4% formaldehyde and 2.5% glutaraldehyde, and then post-fixed in 0.1 M cacodylate buffer (pH 7.2) with 1% osmium tetroxide and1.5% potassium ferrocyanide for 1 h, followed by 1% tannic acid in 0.1 M cacodylate buffer (pH 7.2) for 1 h, finally in 1% uranyl acetate for 1 h. Subsequently, the samples were dehydrated in ethanol, embedded in TAAB 812 resin and polymerized for 24 h at 60 °C. Sections were cut with Reichert Ultracut ultramicrotome and examined with FEI Tecnai 12 Biotwin microscope at 100 kV accelerating voltage. Images were taken with Gatan Orius SC1000 CCD camera.

**Pyruvate dehydrogenase activity Assay**. Pyruvate dehydrogenase (Pdh) enzyme activity assay kit was purchased from Abcam (ab109902). Homogenated heart tissues or cells were extracted for measurement of Pdh activity following the manufacturer's instruction.

**Assessment of β-hydroxylacyl CoA dehydrogenase activity**. The enzymatic activity of β-hydroxylacyl CoA dehydrogenase was measured based on the continuous spectrophotometric rate determination, following the reduction of $Nad^+$ at 340 nm. A final volume of 190 μl with 10 μl heart lysates (10 μg/μl), 160 ml of 50 mM imidazole (pH 7.4, Sigma, 56750) and 20 μl of 1.5 mM Nadh (Sigma, N8129) was pipetted into a 96-well plate. The reaction was initiated by adding 10 ml of 2 mM acetoacetyl CoA (Sigma, A1625), and the absorbance was detected using a kinetic plate reader at a 340 nm wavelength.

**mtDNA copy number**. Total DNA was extracted from myocardium using phenol/chloroform/isoamyl alcohol (25:24:1) followed by isopropanol precipitation. The content of mtDNA was measured by quantifying mitochondrial gene Cox1 (forward 5′ ACTATACTACTACTAACAGACCG-3′, reverse 5′-GGTTCTT TTTTTCCGGAGTA-3′) against nuclear gene cyclophilin A (forward 5′-ACACG CCATAATGGCACTGG-3′, reverse 5′-CAGTCTTGGCAGTGCAGAT-3′) using real-time PCR by the comparative $2^{-\Delta\Delta CT}$ method[46].

**Neonatal rat cardiomyocytes (NRCMs) isolation and culture**. As previously described[39], NRCMs were isolated from 2 days old Sprague Dawley rats using an enzyme solution (30 U/100 ml collagenase A, 100 mg/ml pancreatin, NaCl 116 mM, HEPES 20 mM, $NaH_2PO_4$ 1 mM, glucose 6 mM, KCl 5 mM, $MgSO_4$ 0.8 mM, pH 7.4). NRCMs were cultured in the medium (79.5% DMEM, 19.5% M199; 1% FBS, 1% penicillin-streptomycin, 2.5 μg/ml amphotericin-B and 1 μm bromodeoxyuridine).

**Immunocytochemistry of NRCMs**. NRCMs were transfected with either control siRNA, or rat Erk5 siRNA (100 nM, 5′-AAAGGGTGCGAGCCTATAT-3′, purchased from Sigma) using Lipofectamine Plus reagent according to the manufacturer's instruction (Invitrogen) for 48 h, followed by palmitate acid (PA, 500 μM, Sigma) treatment for 2 h. NRCMs were subject to immunocytochemistry using the primary α-actinin antibody (Sigma, A7811, 1:100) and Mef2a (Abcam, ab32866, 1:100). DAPI was used to visualize their nuclei. Images of 150 visible cells were collected; the fluorescence intensities were measured using Image J software.

**Luciferase reporter assay**. To determine whether Erk5-Mef2 pathway positively regulates Pgc-1α, NRCMs were co-transfected with Erk5 cDNA, Mef2a cDNA and mouse Pgc-1α promoter luciferase reporter (Pgc-1α -Luc, -2112 relative to transcription start site, provided by Dr Daniel P. Kelly, Sanford Burnham Prebys Medical Discovery Institute)[47] for 48 h, GFP cDNA was used as a control. To investigate if palmitate acid upregulates Pgc-1α activity through Erk5, Erk5-knockdown NRCMs were transfected with Pgc-1α promoter luciferase reporter followed by 2 h PA treatment. Thereafter, Pgc-1α luciferase activity was analyzed using luciferase assay kit (Promega).

**Adult rat cardiomyocytes isolation and culture**. Adult rat cardiomyocytes (ARCMs) were isolated as previously described[48] with modification. Eight week old Sprague Dawley rats were administered a lethal dose of pentobarbital (150 mg/kg) with 250 U of heparin. The heart was removed and cannulated for retrograde perfusion through the aorta performed for 5 min with perfusion buffer ($Ca^{2+}$ free Hank's buffered salt solution, 11 mM glucose, 1 mM $MgSO_4$) followed by addition of collagenase (117 U/ml, collagenase type 2, Worthington) and protease (0.175 U/ml protease type XIV, Sigma). The left ventricle was dissected and cut in perfusion buffer containing collagenase and protease, in addition to 0.02 mg/ml trypsin, 0.02 mg/ml DNAse I, 1 mM $CaCl_2$. Digested and filtered ARVMs were resuspended in culture media (1.8 mM $CaCl_2$, 116 mM NaCl, 0.6 mM NaOAc, 1 mM $Na_2PO_4$, 0.8 mM $MgSO_4$, 5.3 mM KCl, 1% BSA, 5mM L-carnitine, 2 mM creatine, 5 mM taurine, 1 mM L-glutamine, 10 mM $Na_2HCO_3$, 10 mM HEPES) with 10 μM blebbistatin and cultured in Geltrex (Gibco) coated plates.

**Adult rat cardiomyocytes transfection with siRNA or infection with adenovirus**. ARCMs were transfected with control siRNA, or rat Erk5 siRNA, rat calpain-1 siRNA (100 nM, siGENOME SMART pool, gene ID #29153, Dharmacon), or rat Gp91phox siRNA (100 nM, #66021) using Lipofectamine Plus reagent according to the manufacturer's instruction (Invitrogen) for 48 h. Erk5-deficient ARCMs were then infected with adenovirus encoding human Pgc-1α (Vector Biolabs) at MOI 25 for 24 h prior to 2-hour PA treatment.

**Fatty acid treatments in adult rat cardiomyocytes**. ARCMs were treated with palmitate (PA, 500 μM, Sigma), stearic acid (SA, 500 μM, Sigma), oleic acid (OA, 500 μM, Sigma) or linoleic acid (LA, 500 μM, Sigma) for 8 h, followed by different investigations. All fatty acid were prepared in 0.5% fatty acid free BSA for conjugation. The complex of fatty acid and BSA (molar ratio 7:1) were applied to mimic pathophysiologic states. To explore the mechanism of Erk5 degradation by PA, ARCMs were pretreated with a combination of cycloheximide (CHX, 10 μg/ml, 4 h) and MG132 (proteasome inhibitor, 10 μM, 4 h) or E-64D (cysteine protease inhibitor, 25 μM, 4 h), followed by PA treatment. To determine which protease is responsible for Erk5 degradation, ARCMs were pretreated with a combination of CHX and calpeptin (calpain inhibitor, 20 μM, 6 h) or cathepsinK inhibitor II (1 μM, 6 h) followed by PA treatment. To assess increased ROS responsible for Erk5 degradation, MDL-28170 (calpain pharmacological inhibitor, 30 μM, 16 h) or Tempol (ROS scavenger, 10 nM, 16 h) were applied to CHX pretreated ARCMs followed by PA treatment. To inhibit the activity of Nadph oxidase, ARCMs were pretreated with apocynin (10 μM) for 1 h followed by application of PA (500 μM, 8 h). To inhibit mitochondria-derived or xanthine oxidase-dependent ROS generation, the mitochondrial targeted antioxidant Coenzyme Q10 (200 μM, Sigma) or xanthine oxidase inhibitor (Allopurinol, 200 μM, Sigma) was applied on ARCMs for 2 h followed by the stimulation of PA.

**HL-1 cell culture**. HL-1, the mouse cardiac muscle cell line (a kind gift from Dr. William C. Claycomb, Louisiana State University), was derived from the AT1 mouse atrial cardiomyocyte tumor lineage[49]. The cells were cultured in Claycomb medium containing 10% FBS, 0.1 mM Norepinephrine, 2 mM L-Glutamine and 100 U Pen-strep. HL-1 cells were transfected with human Erk5 cDNA using FuGene6 following manufacturer's instruction (Promega).

**Chromatin immunoprecipitation**. Chromatin immunoprecipitation (ChIP) was performed using the SimpleChip Plus Enzymatic ChIP kit (Cell Signaling) following manufacturer's instruction. In brief, $2 \times 10^6$ of HL-1 cells were cross-linked with 1% formaldehyde and harvested for the preparation of nuclei suspension. Nuclear membrane was lysed by sonication. Five micrograms of chromatin was used for immunoprecipitation incubated with Mef2a (Abcam). Quantitative PCR was subsequently performed using the following primer sets (binding site 1, -1506 relative to TSS: forward: 5′-CCAGCTCATTTCCTTTACTT-3′, reverse: 5′- CAGACTGAGAGAAGTCACCA-3′; binding site 2, -2868 relative to TSS: forward: 5′-TAGTTGCTCACCAACCTTGG-3′, reverse: 5′-TGA-GAGGCCAGGCTGCACACA-3′; and non-binding intron site: forward: 5′-AGG TATTATTCATTTAATTT-3′, reverse: 5′-ATAGTTTACTGAAGTTTTAC-3′). Data were normalized to the amount of input chromatin. To confirm the binding sequence, 4 ng of PCR fragment was sequenced using ABI 3730 DNA analyzer.

**Calpain-1 cleavage assay**. COS-7 cells (Thermo Fisher Scientific, passage 11) were transfected with Flag-tagged human Erk5 using Lipofectamine LTX (Invitrogen). 24 h post-transfection, cells were then lysed with immunoprecipitation buffer (Tris 50 mM, NaCl 250 mM, 0.25% v/v Triton X-100, and 10% Glycerol; pH 7.4). 500 μg protein was precipitated with 10 μg of anti-Flag antibody (Cell Signaling, 8416) bound to Pierce protein A/G agarose (Thermo Scientific). Precipitated protein was incubated with 1 μg, 2 μg, or 5 μg of human recombinant calpain-1 (Sigma, C6108) in calpain buffer (Tris 50 mM, NaCl 50 mM, 5% glycerol, 0.25 mM $CaCl_2$) at 37 °C for 30 min, followed by immunoblotting using anti-Flag antibody (Sigma, F7425). Furthermore, a cell-free system was used by incubation of human recombinant calpain-1 (2 μg) with 5 pmol or 15 pmol Erk5 (Abcam, ab126913) for 1 min or 5 min in the calpain buffer. The cleavage reaction was analyzed by immunoblotting using anti-Erk5 antibody (Cell Signaling, 3372).

**Lysate preparation and immunoblotting**. Total protein from tissues or cells was obtained with Triton lysis buffer (Tris 20 mM, NaCl 137 mM, EDTA 2 mM, 1% Triton X-100, β-glycerophosphate 25 mM, Na3VO4 1 mM, phenylmethanesulfonylfluoride 1 mM, aprotinin 1.54 μM, leupeptin 21.6 μM, 10% glycerol; pH 7.4). Protein concentration was determined by Bio-Rad assay. Protein extracts (30 μg) were subject to immunoblot analyses with antibodies (all antibodies applied in the current study were used as 1:1000) against Erk5 (Cell Signaling, 3372 or Santa Cruz, sc1286), Erk1/2 (Cell Signaling, 9102), Jnk (Cell Signaling, 9252), p38 (Cell Signaling, 9212), Mef2a (Abcam, ab32866), Mef2c (Abcam, ab64644), Mef2d (Santa Cruz, sc271153), Creb (Cell Signaling, 9197), Ampk (Cell Signaling, 2532), Klf4 (Santa Cruz, sc20691), Mek5 (Santa Cruz, sc10795), cleaved caspase-3 (Cell Signaling, 9661), Ir (Cell Signaling, 3025), Irs1 (Cell Signaling, 2382), calpain-1 (Santa Cruz, sc7531), phosphor-Erk5 (Cell Signaling, 3371), phosphor-Erk1/2 (Cell Signaling, 9101), phosphor-Jnk (Cell Signaling, 9251), phosphor-p38 (Cell Signaling, 9211), phosphor-Ampk (Cell Signaling, 2535), phosphor-Creb (Cell Signaling, 9198), phosphor-Mek5 (Santa Cruz, sc135702), phosphor-Mef2a (Cell

Signaling, 9737), phosphor-Mef2c (Santa Cruz, sc130201), phosphor-Mef2d (Abcam, ab59200), Pgc-1α (Santa Cruz, sc5816), Gp91phox (Santa Cruz, sc130543), or Actin (Abcam, ab20272). Immune-complexes were detected by enhanced chemiluminescence with anti-mouse, anti-rabbit, or anti-goat immunoglobulin-G coupled with horseradish peroxidase as the secondary antibody. Full scan images are available in Supplementary Fig. 14.

**Quantitative real-time polymerase chain reaction**. Total RNA was extracted from ventricular tissues or cells using Trizol (Invitrogen) followed by conversion to cDNA. All primers were purchased from Qiagen. Real-time quantitative PCRs were performed by using SYBR Select PCR Master Mix according to the manufacturer's instruction in the Step One Plus PCR System (Applied Biosystems). The fold change was analyzed using the $2^{-\Delta\Delta CT}$ method[46]. The level of expression was normalized to *Gapdh*.

**Data analysis**. Data were expressed as mean ± SD and analyzed using Student's *t*-test, one-way or two-way ANOVA followed by Bonferroni post hoc tests where appropriate. Comparisons between two groups were performed using Student's *t*-test. *P*-values < 0.05 are considered statistically significant.

**Data availability**. The data that support the findings of this study are available within the article and Supplementary Information or available from the authors upon request.

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

## Acknowledgements

This work was supported by the British Heart Foundation (PG/12/76/29852, PG/14/71/31063, PG/14/70/31039, and FS/15/16/31477) and National Nature Science Foundation of China (81330004).

## Author contributions

W.L., A.R.-V., and S.W. designed and carried out experiments, analyzed results, and wrote the manuscript. S.K., M.Z., M.D.C-.M., J.G., and L.X. performed experiments. A.J. carried out AAV9 virus production and wrote the manuscript. G.D., L.X., and Y.J. helped design experiments and reviewed the manuscript. A.N., G.G., O.J.M., and E.J.C. helped design experiments, interpret results and reviewed the manuscript. W.L. and X.W. designed the study and experiments, interpreted results and wrote the manuscript.

## Additional information

**Competing interests:** The authors declare no competing financial interests.

