## [Peer Review File · Nature Communications]

Reviewers' comments:

Reviewer #1 (Remarks to the Author):

This study demonstrates that loss of ERK5 protein in the heart is a feature of metabolic cardiomyopathy common to several independent models/species. Moreover, cardiac-specific deletion of ERK5 phenocopies the syndrome. Further, data are presented supporting a mechanism in which ROS - induced by free fatty acids through gp91phox - activates calpain-mediated degradation of ERK5. This results in inactivation of an ERK5-MEF2A/D-PGC1alpha cascade, which ultimately results in both mitochondrial dysfunction with impaired ATP generation and decreased numbers of mitochondria. The functional relevance of the pathway is demonstrated by a rescue experiment in which exogenous ERK5 (delivered to the heart by AAV) rescues multiple aspects of the phenotype exhibited by cardiac-specific ERK5 knockout mice.

Strengths of the work include novelty and the quality and richness of the data set. Regarding the latter, the phenotypic characterization was quite detailed, and many interesting aspects were explored. Moreover, this study featured a very effective integration of cell culture and in vivo approaches.

The main weakness of the work in this reviewer's opinion is the mechanistic proof linking calpain to the degradation of ERK5 could be stronger. While knockdown or small molecule inhibition of calpain clearly inhibits ERK5 degradation under pathological conditions, these effects may not necessarily be direct. The data would be strengthened by demonstrating, at a minimum, that calpain cleaves ERK5 directly. This might be accomplished using either recombinant proteins or a cell-based system. In addition, it would be optimal if the investigators could show that an ERK5 mutant that is resistant to calpain cleavage was able to rescue the fatty acid-induced phenotype in cardiomyocytes with ERK5 knockdown - while comparable levels of wild type ERK5 could not. The reviewer understands the challenges inherent in the latter request including (a) difficulties in identifying the calpain cleavage site in ERK5 (suggestions: consider an informatics approach; do mass spec as an alternative); (b) the existence of multiple cleavage sites (not necessarily, but possible); and (c) problems with seeing differential rescue from a cleavage-resistant mutant versus wild type ERK5, if you are only able to achieve marked overexpression. Regarding this latter point, I note that even wild type ERK5 transduced by AAV rescued the phenotype in vivo likely because of overexpression. One way of dealing with this may be to do a knockin of the mutant using CRISPR/Cas9, although you would not be able to do this using primary cardiomyocytes. However, the general point here is that it would be worthwhile to more definitively link calpain with ERK5 because this is the most central and novel aspect of the mechanism you propose.

It is also true that the mechanisms underlying the events downstream of decreased PGC1a on mitochondrial function and number were not explored in this study. For example, why Complex I activity decreases? Or why is mitochondrial number decreased - regarding this, one might suspect decreased mitochondrial biogenesis because of the involvement of PGC1alpha - although increased mitophagy may also be a factor given the mitochondrial abnormalities that were described. While it would be interesting to understand these mechanisms, this reviewer believes that they are not central to the major point of the paper. And can go unexplored in this study, which already includes a substantial amount of data.

A few other more minor points:

What was the mechanism underlying the decreases in MEF2A and MEF2D?

Why in Fig 7b does it appear that you do not rescue beta-hydroxyacyl CoA dehydrogenase activity with siCal and MDL? By not rescue, I am comparing these data with those in Figs 3f and 8i.

I would change your use of the term mitochondrial gene expression programs. There could be confusion with mitochondrial-encoded genes.

Reviewer #2 (Remarks to the Author):

The report on the role of ERK5 in regulating the cardiac response to high fat diet (HFD) presents interesting results, but several general points are not as well considered as possible. The understanding that ERK5 is generally thought to be pro-hypertrophic when activated does not seem to be well considered in this knock out model when examining hypertrophy (see Appari M, et al, Circ Res 2016). Since mice with a 57/Black background are well know to have cardiac dysfunction in response to HFD, the authors do not consider that the ERK5 deletion actually exacerbates impaired cardiac function due to HFD that appears to be present in the data, especially diastolic dysfunction known to occur in HFD C57/black mice, but perhaps is not well powered to indicated for the HFD control flox groups.

Some of the rationale and background information is not quite accurate and potentially misleading to readers who are not expert in the field. Expansion of the endogenous triglyceride pool in the heart is not in itself toxic, as is generally now known, but rather changes in the endogenous TG pool are reflective of altered lipid metabolism with the potential for other, non TG pool lipid species to be altered and exert deleterious biochemical effects. TG does not induce ROS and mitochondrial dysfunction as the authors incorrectly state. Furthermore, the authors offer an inaccurate and overly simplified version of metabolic shifts in the pathological heart that ignore the well established energetic deficit induced by inefficient carbohydrate oxidation and reduced fat oxidation. However, this pathological model is very different than the etiology of the diabetic heart in which fatty acid oxidation is very high to point of being restrictive to metabolic plasticity and glucose metabolism. It is not all the consequence of toxic lipid intermediates and diabetic hearts do not show reduced fat oxidation, so the authors are confusing the etiologies and the topic in general.

There are some general concerns regarding the experimental model. The use of neonatal rat cardiomyocytes is not an appropriate model for examining the effects of fatty acid exposure due to the low mitochondrial density and low LCFA oxidation capacity and rates of the NRCM. Apart from the complete lack of ATP demand of the unloaded, inactive cardiomyocyte, the NRCM does not represent metabolic stress responses of an adult cardiomyocyte and cannot be extrapolated to an understanding of the adult mouse heart results.

Specific Comments

1. The authors state the use of an ERK5-CKO model to "mimic" the ErK5 reduction they report in insulin resistant hearts. However, the knock out does not mimic reduction. This is a much different level of expression that is not represented by the KO.
2. The loading protein control should be calcineurin or actin and not GAPDH since glycolytic capacity has the potential to change in response to these metabolic protocols.
3. In figure 1, the bar graph is somewhat confusing and should simply represent the values from each group as single bars without the white and black sections.

4. The results text is a bit confusing regarding data shown in Figure 1. If n=6 for all groups, then please show n=6 bands for each experimental group and not n=3.

4. As indicated in the general comments regarding several data sets, Figure 2g, in particular, shows results that indicate a profound response of the Flox HFD group dPW that is not addressed by the authors.

5. Not all ceramides are deleterious, but rather specific, long chain species have been implicated in cardiac lipid toxicity. Thus, it would be helpful to know which species of ceramides are changing rather than just total ceramide that holds much less meaning.

Reviewer #3 (Remarks to the Author):

In the present study authors describe a novel molecular mechanism by which ERK5 regulates PGC1- α during metabolic stress. ERK5 is a protein kinase involved in different cellular processes and its ablation causes aberrant mitochondrial gene expression and subsequent cardiac dysfunction upon metabolic stress. ERK5 is selectively upregulated in different animal models of metabolic stress, including 16week old mice fed with High Fat Diet (HFD), whereas HFD mice at 25w have decreased levels of ERK5. Thus, authors use ERK5 cardiac knockout mice (ERK5 cko) and subject them to chow or high fat diet, showing that ERK5 loss combined with HFD leads to mitochondrial dysfunction and cardiomyopathy, prevented when ERK5 levels were restored. Mechanistically, they show that cardiac ERK5 is degraded in a number of "dysmetabolic" conditions by gp91phox activation of calpain-1. This degradation reduces ERK5/MEF2 signalling upstream of PGC-1 α and hence the mitochondrial transcriptional program.

In general, this is an interesting paper showing that excess dietary fat leads to ERK5 reduction and hence to loss of mitochondrial function. However, in the current version the authors fail to demonstrate that ERK5 effects are mediated by PGC1 α . While they show that ERK5 ablation reduces a number of transcripts controlled by PGC1 α , there is no formal proof that PGC1 α activation can compensate for ERK5 degradation. In addition, there are certain inconsistencies on the proposed mechanism. For example, how do authors explain that levels of MEF2C are not changed upon HFD? ERK5 is known to regulate MEF2C and authors show in figure 6a that MEF2C specifically changes upon ERK5 silencing. Finally, a major phenotype that appears to be overlooked (and might explain the phenotype, perhaps even more than PGC1 α), is the association between mitochondria and lipid droplets (EMs in Figure 3). This phenotype occurs only in ERK5 cKO hearts under HFD, suggesting that ERK5 negatively regulates FA utilization by mitochondria, or more generally lipid droplets maturation/degradation/association with mitochondria. In conclusion, authors shall

1. Verify if the phenotype is mediated by PGC1 α . This can be done in vitro or (better) in vivo, by checking if constitutively active PGC1 α compensates ERK5 loss
2. Explore the role of ERK5 in mitochondrial FA utilization and/or LD maturation. For example, the ability of ERK5-deficient mitochondria to respire using FA shall be compared to the WT counterparts; the association between LD and mitochondria shall be quantified and correlated to the FA-driven respiration; whether mitochondrial FA utilization control by ERK5 depends on PGC1 α (very likely) shall also be tested, to verify if PGC1 α reactivation corrects ERK deficiency.

Minor comments

Abstract, line 8: "free fatty acid" and not "acids"

Abstract, line 9: the use of AAV9 does not prevent ERK5 loss, is restoring ERK5 levels.

Introduction, second paragraph, last sentence: " a key question...is that the mechanism....remains unknown". Is not syntactically right.

Introduction, third paragraph, first sentence: "transcriptional" and not "transcription" activity.

Discussion, first sentence: "the" unique phenomenon that and not "a" unique

Page 4, 6th line: skeletal and not skeleton muscle

Page 6, 1st line: "concomitantly with" and not "concomitant with"

Discussion, second paragraph: "AMPK altered in the present study" and not "AMPK was altered from the present study"

1a: miss the (C-ter)

9: typo mistake "OXPHOX" to be replaced by "OXPHOS"

RESPONSE TO THE REVIEWERS' COMMENTS

First, we would like to thank the editor and the reviewers for their constructive comments, which have helped us tremendously in improving our manuscript. We have conducted new experiments as suggested by the editor and reviewers and have added new data to this revised manuscript. We have written verbatim the comments of the editor and reviewers followed by our response, and indicate where changes have been made in the manuscript. Due to changes in the text and in references, please refer to page number, figure number, reference number in revised manuscript and online data supplement.

Reviewer 1

This study demonstrates that loss of ERK5 protein in the heart is a feature of metabolic cardiomyopathy common to several independent models/species. Moreover, cardiac-specific deletion of ERK5 phenocopies the syndrome. Further, data are presented supporting a mechanism in which ROS - induced by free fatty acids through gp91phox - activates calpain-mediated degradation of ERK5. This results in inactivation of an ERK5-MEF2A/D-PGC1alpha cascade, which ultimately results in both mitochondrial dysfunction with impaired ATP generation and decreased numbers of mitochondria. The functional relevance of the pathway is demonstrated by a rescue experiment in which exogenous ERK5 (delivered to the heart by AAV) rescues multiple aspects of the phenotype exhibited by cardiac-specific ERK5 knockout mice.

Strengths of the work include novelty and the quality and richness of the data set. Regarding the latter, the phenotypic characterization was quite detailed, and many interesting aspects were explored. Moreover, this study featured a very effective integration of cell culture and in vivo approaches.

Comment 1: The main weakness of the work in this reviewer's opinion is the mechanistic proof linking calpain to the degradation of ERK5 could be stronger. While knockdown or small molecule inhibition of calpain clearly inhibits ERK5 degradation under pathological conditions, these effects may not necessarily be direct. The data would be strengthened by demonstrating, at a minimum, that calpain cleaves ERK5 directly. This might be accomplished using either recombinant proteins or a cell-based system. In addition, it would optimal if the investigators could show that an ERK5 mutant that is resistant to calpain cleavage was able to rescue the fatty acid-induced phenotype in cardiomyocytes with ERK5 knockdown, while comparable levels of wild type ERK5 could not. The reviewer understands the challenges inherent in the latter request including (a) difficulties in identifying the calpain cleavage site in ERK5 (suggestions: consider an informatics approach; do mass spec as an alternative); (b) the existence of multiple cleavage sites (not necessarily, but possible); and (c) problems with seeing differential rescue from a cleavage-resistant mutant versus wild type ERK5, if you are only able to achieve marked overexpression. Regarding this latter point, I note that even wild type ERK5 transduced by AAV rescued the phenotype in vivo likely because of overexpression. One way of dealing with this may be to do a knockin of the

mutant using CRISPR/Cas9, although you would not be able to do this using primary cardiomyocytes. However, the general point here is that it would be worthwhile to more definitively link calpain with ERK5 because this is the most central and novel aspect of the mechanism you propose.

Response: We are very grateful for this critical comment and have performed experiments to provide a direct link of ERK5 degradation by calpain-1. First of all, we used an informatics approach (*in silico* programmes GPS-CCD 1.0, CaMPDB, and CALPCLEAV) to search putative ERK5 cleavage sites by calpain-1 and found out multiple potential cleavage sites, most of which are enriched between amino acid 290 to 600. We then adopted a cell-based system to transfect Flag-tagged human ERK5 in COS-7 cells. In the presence of 10mM CaCl₂, the cell lysates were incubated with human recombinant calpain-1 (1µg, 2µg and 5µg) for 1 hour, and then subject to immunoblot analysis using anti-Flag antibody. As anticipated, multiple bands ranging from 55KD to 25KD were captured, indicating several calpain-1 cleavage sites are present in the ERK5 sequence. This observation provides direct evidence of ERK5 degradation by calpain-1 and is consistent with our previous data that after palmitate treatment ERK5 was not detected by both antibodies targeting either its N-terminus or C-terminus. Due to the existence of multiple cleavage sites, an approach using calpain-resistant ERK5 mutant to rescue the fatty acid-induced phenotype seems unlikely. Of note, using ERK5 overexpression to rescue fatty acid-induced phenotype is duration-dependent. If FFA stress becomes persistent, eventually overexpressed ERK5 would be degraded. Therefore, AAV9-delivery of ERK5 restoration is to provide “proof of concept” evidence showing ERK5 protective role in FFA stressed hearts. From clinical point of view, prevention of ERK5 degradation would open a new treatment avenue.

The new data has been added in the revised manuscript.

Result section: page 6, lines 34-39.

New sentences added in to describe these experiments and obtained result.

New data added in **Supplementary Fig. 8** to present aforementioned result, and new figure legend provided.

Methods: page 14, lines 6-12.

Comment 2: It is also true that the mechanisms underlying the events downstream of decreased PGC1a on mitochondrial function and number were not explored in this study For example, why Complex I activity decreases ? Or why is mitochondrial number decreased – regarding this, one might suspect decreased mitochondrial biogenesis because of the involvement of PGC1alpha – although increased mitophagy may also be a factor given the mitochondrial abnormalities that were described. While it would be interesting to understand these mechanisms, this reviewer believes that they are not central to the major point of the paper. And can go unexplored in this study, which already includes a substantial amount of data.

Response: We thank the reviewer for this comment and giving us the opportunity to clarify this point. The primary role of the mitochondria is to generate ATP through oxidative phosphorylation (OXPHOS). As a major entry point for most electrons into the respiratory chain, complex I

(NADH:ubiquinone oxidoreductase) is the largest complex with multi-step assembly process and the rate-limiting step of the mitochondrial OXPHOS system, playing an essential role in energy production. Defective complex I activity is a critical determinant of impaired OXPHOS capacity. The most frequently observed OXPHOS disorders are associated with complex I dysfunction, independent of any other dysfunction in the electron transport chain (Irwin et al., *Int J Biochem Cell Biol.* 2013,45:34-40), possibly due to its proximity and susceptibility to mitochondria-generated ROS. (1) In our study, the loss of ERK5 caused a significant down-regulation of PGC-1 α -dependent transcription programmes in response to FFA stress. As a result, a wide array of mitochondrial function genes was decreased, including important complex I subunits, *ndufa7*, *ndufs8*, *nd6*, and mitochondrial biogenesis genes, such as *tfb1m*, *tfb2m* and *mfn2*. Those reductions gave rise to impaired complex I activity and decreased mitochondrial numbers, respectively (**Fig. 3a-d**). New sentences have been added to discuss impaired complex I activity in **page 8, line 48 to page 9, line 2**. (2) We have examined key factors of mitophagy, which is important for regulating mitochondrial dynamics in number, size, shape and distribution in response to extracellular cues. We used adult rat cardiomyocytes as a cellular model. In the presence of lysosomal inhibitor chloroquine, ERK5 knockdown did not affect rapamycin-induced increased conversion of LC3-I to LC3-II and enhanced PINK1 expression (see attached Figure). This data suggests that ERK5 may not exert a direct impact on mitophagy. However, to confirm this notion, further investigations on mitophagy (such as mitokeima assay) and the assessment of mitochondrial fusion/fission will be needed. Finally, we appreciate that the reviewer acknowledged that a considerable investigation into mitophagy is beyond the current study and can go unexplored in this manuscript; therefore our initial attempt on mitophagy is herein presented.

Figure legend: Adult rat cardiomyocytes were pre-treated with chloroquine (3 μ M) for 2 hours, followed by incubation of rapamycin (5 μ M) for 2 hours. Thereafter, Western blot analysis showed comparable levels of PINK1 expression and conversion of LC3-I to LC3-II regardless of ERK5 presence or absence.

Minor points

1. What was the mechanism underlying the decreases in MEF2A and MEF2D?

Response: MEF2 family members are self-regulated transcription factors. Previous studies have been demonstrated that MEF2 transcription is subject to regulation by its protein products, depending on signalling regulation that

influences MEF2 factor transcriptional activity (Ramachandran et al., J Bio Chem., 2008, 283:10318-10329; Cripps et al., De Bio., 2004, 267:536-547). In line with this premise, we believe in current study that the loss of ERK5 resulted in decreased MEF2 phosphorylation and subsequent reduced transcriptional activity, leading to a diminution in its expression at both mRNA and protein level. Above mentioned references have been added in the main text accordingly as refs 28, 29 to explain this observation. Reduced expression of MEF2A/D at mRNA and protein level has been presented in **Supplementary Fig. 7**. It is worth pointing out that we did not observe changes in the phosphorylation level and total expression of MEF2C (**Fig 6a**). Unlike MEF2A and MEF2D which are specific downstream of ERK5, MEF2C is subject to regulation by ERK5 as well as p38. This could explain why its expression and phosphorylation remained unchanged in HFD-CKO hearts. The sentences have been added to clarify this point in **the Discussion (page 7, lines 53-54)**.

2. Why in Fig 7b does it appear that you do not rescue beta-hydroxyacyl CoA dehydrogenase activity with siCal and MDL? By not rescue, I am comparing these data with those in Figs 3f and 8i.

Response: We appreciate that the reviewer spotted this mistake. To better measure metabolic responses upon palmitate stimulation, we have conducted cellular experiments in adult rat cardiomyocytes. In the new set of data, calpain-1 knockdown or MDL-28170 was able to rescue PA-induced the reduction of β -hydroxylacyl CoA dehydrogenase activity that was presented in **Fig. 7b**.

3. I would change your use of the term mitochondrial gene expression programs. There could be confusion with mitochondrial-encoded genes.

Response: We totally agree with this suggestion. This term has been changed to “mitochondrial function-related gene programme” throughout the whole manuscript.

Reviewer 2

Comment 1: The report on the role of ERK5 in regulating the cardiac response to high fat diet (HFD) presents interesting results, but several general points are not as well considered as possible. The understanding that ERK5 is generally thought to be pro-hypertrophic when activated does not seem to be well considered in this knock out model when examining hypertrophy (see Appari M, et al, Circ Res 2016)

Response: Thank the reviewer for raising this important point and giving us this opportunity to clarify the role of ERK5 in hypertrophy. In our previous study (Kimura et al., Circ Res., 2010,106: 961-970), we reported that the mice defective of cardiac ERK5 failed to resist pressure overload, manifesting cardiac dysfunction and less hypertrophy, succumbing to heart failure due to cumulative cardiomyocyte death. In consistent, we also observed less hypertrophy induced by HFD in ERK5-CKO mice in the current study (**Fig. 2g**,

Supplementary Fig. 4, Supplementary Fig. 5). A recent study by Appari et al. provided further evidence linking ERK5 to hypertrophy (**new ref 32 is added**). In the CTRP9 knock-out mouse model, they showed reduced level of ERK5 during TAC compared with wild-type mice, while CTRP9 overexpression entailed increased ERK5 activation in response. Inhibition of ERK5 by a dominant negative MEK5 mutant or by the ERK5/MEK5 inhibitor BIX02189 blunted CTRP9 triggered hypertrophy. Collectively, these findings duly demonstrate the requirement of ERK5 in development of hypertrophy. However, future study will be needed to investigate ERK5 overexpression mouse model, therefore directly establishing pro-hypertrophic status of ERK5 in the heart. New sentences have been added in to discuss this point in **the Discussion (page 9, lines 3-11)**.

Comment 2: Since mice with a 57/Black background are well know to have cardiac dysfunction in response to HFD, the authors do not consider that the ERK5 deletion actually exacerbates impaired cardiac function due to HFD that appears to be present in the data, especially diastolic dysfunction known to occur in HFD C57/black mice, but perhaps is not well powered to indicated for the HFD control flox groups.

Response: We thank the reviewer for raising this point and giving us this opportunity to make a clarification. Previous studies showed that 25 weeks of HFD caused impaired cardiac function in C57BL/6 black mice (Pavan et al., J Clin Invest. 2012, 122: 1109-1118). However, in our study cardiac ERK5 deletion predisposed the mice to dampened contractility and marked cardiac mitochondrial abnormalities after 16 weeks of HFD, whilst at this feeding time point, Flox mice with C57 background remained normal cardiac function and ERK5 expression was preserved intact in ERK5-Flox hearts (**the Results section, page 4, lines 20-21; Fig. 2g and Supplementary Fig. 1c**). Therefore, we reckon that duration of HFD is likely an explanation for this discrepancy between ERK5-CKO mice and C57BL/6 mice. Of note, all our data has been evaluated by power calculations.

Comment 3: Some of the rationale and background information is not quite accurate and potentially misleading to readers who are not expert in the field. Expansion of the endogenous triglyceride pool in the heart is not in itself toxic , as is generally now known, but rather changes in the endogenous TG pool are reflective of altered lipid metabolism with the potential for other, non TG pool lipid species to be altered and exert deleterious biochemical effects. TG does not induce ROS and mitochondrial dysfunction as the authors incorrectly state. Furthermore, the authors offer an inaccurate and overly simplified version of metabolic shifts in the pathological heart that ignore the well established energetic deficit induced by inefficient carbohydrate oxidation and reduced fat oxidation. However, this pathological model is very different than the etiology of the diabetic heart in which fatty acid oxidation is very high to point of being restrictive to metabolic plasticity and glucose metabolism. It is not all the consequence of toxic lipid intermediates and diabetic hearts do not show reduced fat oxidation, so the authors are confusing the etiologies and the topic in general.

Response: We are most grateful to the reviewer for this important comment. We acknowledge that expansion of TG pool is not toxic and have carefully checked the text to be accurate. It is well documented that the diabetic heart relies almost on FFA utilization with a decrease in glucose utilization via the Randle cycle. However, as metabolic stress becomes more prolonged, mitochondrial uncoupling contributes to increasing oxygen consumption without a concomitant increase in ATP production, which causes decreased cardiac efficiency in the diabetic heart. Furthermore, oxidative damage co-existence with necrosis and apoptosis gives rise to the onset of heart failure. At this stage both FFA oxidation and glucose utilization are blunted. With regard to this pathological progress from adaptive stage to decompensated stage, we have described it in **the Introduction section (page 3, lines 11-25)**. Our ERK5-CKO mice subject to 16 weeks of HFD represent as a cardiomyopathy model in its failing stage induced by prolonged metabolic stress, therefore we observed energetic deficit caused by inefficient glucose utilization and reduced fat oxidation.

Comment 4: There are some general concerns regarding the experimental model. The use of neonatal rat cardiomyocytes is not an appropriate model for examining the effects of fatty acid exposure due to the low mitochondrial density and low LCFA oxidation capacity and rates of the NRCM . Apart from the complete lack of ATP demand of the unloaded, inactive cardiomyocyte, the NRCM does not represent metabolic stress responses of an adult cardiomyocyte and cannot be extrapolated to an understanding of the adult mouse heart results.

Response: We truly appreciate this comment. In the previous version, we used NRCMs after 10 days culture. Under this condition, we were able to study ERK5 signalling pathway and functional differences attributable to the loss of ERK5, although it is true that NRCMs have less LCFA oxidation capacity and low mitochondrial density. In the revised manuscript, we have carried out a series of cellular studies in adult rat cardiomyocytes (ARCMs). We greatly thank the reviewer for this suggestion, which significantly improves the quality of our manuscript and new cellular data can be extrapolated to an understanding of the adult mouse heart results. The new set of data is presented as **Fig. 6f-l, Fig. 7, Supplementary Fig. 7, 9 and 11**. Method for preparing adult rat cardiomyocytes has been included in **the Methods section, page13, lines 9-26**. Text and legends have been changed accordingly.

Specific Comments

1. The authors state the use of an ERK5-CKO model to “mimic” the Erk5 reduction they report in insulin resistant hearts. However, the knock out does not mimic reduction. This is a much different level of expression that is not represented by the KO.

Response: Thanks for this comment. We have changed the wording as “we used ERK5-CKO mice to investigate the role of ERK5 in the obese/diabetic hearts.” Please see **page 4, line 19**.

2. The loading protein control should be calquestrin or actin and not GAPDH since glycolytic capacity has the potential to change in response to these metabolic protocols.

Response: We thank this comment. Following this suggestion, protein loading control for all Western blot analysis has been changed to actin. New data is presented in **Fig. 1, 6, 7, and Supplementary Fig. 1, 7, 10, 12,** and legends have been changed accordingly.

3. In figure 1, the bar graph is somewhat confusing and should simply represent the values from each group as single bars without the white and black sections. The results text is a bit confusing regarding data shown in Figure 1. If n=6 for all groups, then please show n=6 bands for each experimental group and not n=3.

Response: Thanks for the comments. We have changed the graph bars as single bars for each group (**Fig. 1a**). For the 2nd point, we have performed experiments and statistical analysis of 6 samples per group; whereas Western blotting is a representative image for loading 3 samples/each group in one gel. We have stated this clearly in new figure legend as “n=6 animals per group for statistic analysis, and Western blotting is a representing image of 3 samples per group”.

4. As indicated in the general comments regarding several data sets, Figure 2g, in particular, shows results that indicate a profound response of the Flox HFD group for increased dPW that is not addressed by the authors.

Response: We thank for this comment. In the revised manuscript, we increased N number to 12 mice per group and provided new data for dPW (**Fig. 2g**) and n=10 mice per group for cross sectional area (**Supplementary Fig. 4**), which showed less hypertrophy in HFD-CKO group compared with HFD-Flox group. We believe that although ERK5-CKO developed less hypertrophy, but substantial cell death is a culprit of cardiac dysfunction and subsequent heart failure. A new paragraph to discuss the link of ERK5 with hypertrophy in response to diet stress is included in **the Discussion section page 9, lines 3-11**. Figure legends have been changed accordingly.

5. Not all ceramides are deleterious, but rather specific, long chain species have been implicated in cardiac lipid toxicity. Thus, it would be helpful to know which species of ceramides are changing rather than just total ceramide that holds much less meaning.

Response: We truly appreciate the reviewer for raising this point. We have made significant efforts to address this issue. Ultraperformance liquid chromatography coupled to a triple quadrupole mass spectrometer with electrospray ionisation was used to analyze ceramide species. Our new data has shown a differential profile of ceramide species in ERK5-CKO hearts versus ERK5-Flox hearts. ERK5-CKO mice despite of diet choice displayed increased levels of multiple ceramide species. The level of 6 species was

increased in chow-CKO compared with chow-Flox hearts, and the level of 8 species was higher in HFD-CKO compared with HFD-Flox hearts. The amount of CER[N(24)S(16)] and CER[N(24)S(17)] were augmented in HFD-CKO hearts compared with HFD-Flox and chow-CKO hearts. To substantiate whether changes in these species, particularly N(24)S(16) and N(24)S(17), would contribute to cardiac lipid toxicity, further investigations on the role of individual ceramide synthases are needed. In addition to ceramides, we have also detected diacylglycerol (DAG) and found its level was considerably increased in ERK5-CKO hearts with HFD. Collectively, we believe that changes in selective ceramide species and elevated DAG are likely deleterious to ERK5-CKO hearts.

The new data has been added in the revised manuscript.

Result section: page 5, lines 18-27.

New sentences added in to describe these experiments and obtained results.

New data added in **Fig. 4c and 4d** to present aforementioned results, and new figure legend provided.

Methods: page 11, line 44 to page 12, line 2.

Discussion section: page 8, lines 44-48, sentences added to discuss the new data.

Reviewer 3

In the present study authors describe a novel molecular mechanism by which ERK5 regulates PGC1- α during metabolic stress. ERK5 is a protein kinase involved in different cellular processes and its ablation causes aberrant mitochondrial gene expression and subsequent cardiac dysfunction upon metabolic stress. ERK5 is selectively upregulated in different animal models of metabolic stress, including 16week old mice fed with High Fat Diet (HFD), whereas HFD mice at 25w have decreased levels of ERK5. Thus, authors use ERK5 cardiac knockout mice (ERK5 cko) and subject them to chow or high fat diet, showing that ERK5 loss combined with HFD leads to mitochondrial dysfunction and cardiomyopathy, prevented when ERK5 levels were restored. Mechanistically, they show that cardiac ERK5 is degraded in a number of “dysmetabolic” conditions by gp91phox activation of calpain-1. This degradation reduces ERK5/MEF2 signalling upstream of PGC-1 α and hence the mitochondrial transcriptional program.

In general, this is an interesting paper showing that excess dietary fat leads to ERK5 reduction and hence to loss of mitochondrial function. However, in the current version the authors fail to demonstrate that ERK5 effects are mediated by PGC1 α . While they show that ERK5 ablation reduces a number of transcripts controlled by PGC1 α , there is no formal proof that PGC1 α activation can compensate for ERK5 degradation. In addition, there are certain inconsistencies on the proposed mechanism. For example, how do authors explain that levels of MEF2C are not changed upon HFD? ERK5 is known to regulate MEF2C and authors show in figure 6a that MEF2C specifically changes upon ERK5 silencing. Finally, a major phenotype that appears to be overlooked (and might explain the phenotype, perhaps even more than PGC1 α), is the association between mitochondria and lipid droplets (EMs in Figure 3). This phenotype occurs only in ERK5 CKO hearts under

HFD, suggesting that ERK5 negatively regulates FA utilization by mitochondria, or more generally lipid droplets maturation, degradation, association with mitochondria. In conclusion, authors shall:

Comment 1: Verify if the phenotype is mediated by PGC-1 α . This can be done in vitro or (better) in vivo, by checking if constitutively active PGC1 α compensates ERK5 loss.

Response: We are grateful to the reviewer for this critical point. PGC-1 α is a master transcriptional coactivator, which dictates widespread gene expression related to mitochondrial biogenesis, fuel generation and oxidative phosphorylation. Because of its importance in inducing a core program of mitochondrial function, overexpression of PGC-1 α in the adult heart led to a modest increase in mitochondrial number, derangements of mitochondrial ultrastructure, and development of cardiomyopathy (Russell et al., *Circ Res.* 2004; 94: 525-533). These results together with findings in PGC-1 α knockout model suggest that optimal level of PGC-1 α is critical for maintaining normal mitochondrial function and metabolic homeostasis (Arany. et al., *Proc Natl Acad Sci USA* 2006; 103: 10086-1009). Over the past 10 years with more than 1000 articles since the discovery of PGC-1 α , we have learnt that PGC-1 α appears to be harmful in liver and pancreas, potentially beneficial in muscle and fat. Given multisystem functions of PGC-1 α , it is therefore that PGC-1 α might not be an ideal therapeutic target for treating cardiac disease, and identifying regulators of PGC-1 α could thus prove useful. Bearing this in mind, we chose a cellular model as overexpressing PGC-1 α in adult rat cardiomyocytes (ARCMs) with ERK5 knockdown. This approach could provide experimental evidence whether PGC-1 α was able to rescue detriments caused by ERK5 deficit. Indeed, we observed normal levels of key mitochondrial genes, such as *ppargc1a*, *cpt1a*, *mcd*, *acadm* and *acaa2* after 2h palmitate treatment (**Supplementary Fig. 11a**). In addition, fuel oxidation measured using acetoacetyl CoA and pyruvate as substrates was improved along with PGC-1 α overexpression (**Supplementary Fig. 11b**). These results corroborate that PGC-1 α is the downstream of ERK5 pathway in cardiomyocytes. From the standpoint of drug development, the identification of approaches to prevent ERK5 degradation rather than PGC-1 α overexpression holds promise for treating metabolic cardiomyopathy.

The new data has been added in the revised manuscript.

Result section: page 7, lines 1-4.

New sentences added in to describe these experiments and obtained results.

New data added in **Supplementary Fig. 11** to present aforementioned results, and new figure legend provided.

Methods: page 13, lines 25-26.

Discussion section: page 8, lines 5-7, sentences added to discuss the new data.

Comment 2: Explore the role of ERK5 in mitochondrial FA utilization and/or LD maturation. For example, the ability of ERK5-deficient mitochondria to respire using FA shall be compared to the WT counterparts; the association between LD and mitochondria shall be quantified and correlated to the FA-driven respiration; whether mitochondrial FA utilization control by ERK5

depends on PGC-1 α (very likely) shall also be tested, to verify if PGC-1 α reactivation corrects ERK deficiency.

Response: We thank the reviewer for this important comment. (1) Further to our previous data measuring electron transport chain (ETC) complex activity and ATP production in cardiac muscle fibers, in the revision we have determined respiration of complex I activity in mitochondria homogenates using fatty acid (palmitoylcarnitine) as a substrate. As anticipated, ERK5-deficient mitochondria prepared from HFD fed mice showed a reduction in FA-driven respiration (**Fig. 3f**) compared to their wild-type counterparts. These results provide direct evidence of ERK5-deficient mitochondria in relation with FA-driven respiration. (2) We have qualified the ratio of lipid droplets density and this data is included in **Fig. 3c**. (3) We also detected mRNA expression of *dgat1* and *atgl*. The former is an enzyme for the formation of triglycerides (TAG) from diacylglycerol (DAG), and the latter is an enzyme for TAG hydrolysis. For both enzymes, we observed an HFD-induced increased mRNA expression to a comparable level in ERK5-CKO and ERK5-Flox hearts (**Fig. 5b**). It will be necessary to assess the activities of these enzymes in future studies to further understand whether ERK5 has an impact on lipid maturation. (4) Regarding whether mitochondrial FA utilization controlled by ERK5 depends on PGC-1 α , and to verify if PGC-1 α reactivation corrects ERK deficiency, see response for comment 1.

The new data has been added in the revised manuscript.

Result section: page 4, lines 46-47; page 5, lines 5-8.

New sentences added in to describe these experiments and obtained results.

New data added in **Fig. 3c, 3f, 5b** to present aforementioned results, and new figure legend provided.

Methods: page 11, lines 25-33.

Discussion section: page 8, lines 5-7; 44-45, sentences added to discuss the new data.

Addition comment: For example, how do authors explain that levels of MEF2C are not changed upon HFD? ERK5 is known to regulate MEF2C and authors show in figure 6a that MEF2C specifically changes upon ERK5 silencing.

Response: We thank the reviewer for this comment. We have repeated several times of this experiment and would like to clarify that we did not observe changes in the phosphorylation level and total expression of MEF2C. New representative Western blotting images have been provided in **Fig 6a**. Unlike MEF2A and MEF2D, which are specific downstream of ERK5, MEF2C is subject to regulation by ERK5 as well as p38. This could explain why its expression and phosphorylation remained unchanged in HFD-CKO hearts. The sentences have been added to clarify this point in **the Discussion (page 7, lines 53-54)**.

Minor comments

Comment 1: Abstract, line 8: “free fatty acid” and not “acids”

Response: The text is changed to “free fatty acid”.

Comment 2: Abstract, line 9: the use of AAV9 does not prevent ERK5 loss, is restoring ERK5 levels.

Response: The original sentence in abstract line 9 was intended to describe “prevention of ERK5 degradation by knockdown of gp91phox or calpain-1 or inhibiting its activity”. To make it clear, the sentence has been amended as “whereas the prevention of ERK5 loss by blocking gp91phox or calpain-1 rescued mitochondrial functions”.

Comment 3: Introduction, second paragraph, last sentence: “a key question...is that the mechanism....remains unknown”. Is not syntactically right.

Response: The text is changed to “.....remains largely unknown” in page 3, line 26.

Comment 4: Introduction, third paragraph, first sentence: “transcriptional” and not “transcription” activity.

Response: The text is changed to “transcriptional activity” in page 3, line 28.

Comment 5: Discussion, first sentence: “the” unique phenomenon that and not “a” unique

Response: The text is changed to “the unique phenomenon” in page 7, line 22.

Comment 6: Page 4, 6th line: skeletal and not skeleton muscle

Page 6, 1st line: “concomitantly with” and not “concomitant with”

Discussion, second paragraph: “AMPK altered in the present study” and not “AMPK was altered from the present study”

1a: miss the (C-ter)

9: typo mistake “OXPHOX” to be replaced by “OXPHOS”

Response: All typo errors have been corrected in text or figures accordingly.

Reviewers' comments:

Reviewer #1 (Remarks to the Author):

The revision is improved. While I pointed out several issues in my initial review, I focused on strengthening the data showing showing that calpain cleaves ERK5 because this is central to mechanism.

The authors first used informatics to identify multiple potential calpain cleavage sites in ERK5. Then, they expressed FLAG-tagged ERK5 in cells, and incubated cell lysates with recombinant calpain in the presence of Ca²⁺. Immunoblotting for FLAG revealed multiple bands smaller than full length FLAG-ERK5 consistent with calpain-mediated cleavage.

I agree that it is CONSISTENT with calpain-mediated cleavage of ERK5 (as are the inhibitor studies presented in the Results), but the problem with this approach is that it does not show that calpain is what is the protease that is actually cleaving ERK5. In theory, calpain could be cleaving and activating another protease, which is then cleaving ERK5.

The most definitive way to show direct cleavage of ERK5 by calpain would have been to use a cell-free system: recombinant ERK5, recombinant calpain, Ca²⁺, and buffer. Then do an immunoblot.

But, in my initial comments to Authors, I said that it would be acceptable to use either an in vitro (cell-free) approach as described above, or a cell-based system. But, if you used a cell-based system, I had assumed that you would try to purify the FLAG-ERK5 by immunoprecipitation before incubating it with the recombinant calpain and Ca²⁺. The way you did it includes the entire cell lysate. An IP step would have lessened concerns that another protease is involved in cleaving ERK5.

But, there is a third option. I would be willing to accept that calpain might be directly cleaving ERK5 if you could (a) show a diagram of the ERK5 protein indicating where the potential calpain cleavage sites are (you should probably include this in Supp Fig 8 anyway because it strengthens your case); and (b) definitively identify some of the cleaved bands on the FLAG immunoblot as corresponding to some of the predicted calpain cleavage fragments based on your informatics analysis.

So, two summarize: three options: (a) the cell-free approach; (b) the approach you used but adding IP:FLAG before incubating with calpain and Ca²⁺; or (c) use with your present data but identify some of your bands as corresponding to predicted calpain cleavage fragments based on your informatics.

Reviewer #2 (Remarks to the Author):

The authors have responded constructively with new data and importantly improved experimental protocols. A few minor weaknesses do remain, however.

The authors rely exclusively on inference in implicating DAG and ceramides as lipotoxic with no direct evidence that either play a role in the dysfunction and pathogenesis of the cardiomyopathy that they observe. A statements should be included to clarify this. I would also contend that many things change, both up and down, in hearts subjected to high fat diet and nutrient overload, all of which are part of a compensatory response to metabolic stress. It would be good if the authors can address the approach, conceptually, because if you take something out of the heart and the heart performs poorly, then you put it back in and it alleviates the dysfunction, it merely proves that all of the components serve a purpose. Does the work anything more than one of many components that regulate the

cardiac response to stress?

The abstract states that "Mechanistic studies revealed..." This qualifying term should be left to the readership to determine how mechanistic the study is. Alternative text should replace this opinionated adjective – perhaps "The current study.." or "The protocols revealed..." as suggestions and examples – but, of course, to be determined by the authors.

A question regards the authors' meaning and intent of the text on page 8, 2nd paragraph, line 16, starting "When the heart works at a compensated stage against metabolic stress.." This is difficult to understand. Are the authors saying something about the actual mechanical work of the heart against a change in afterload? Are they talking instead about a metabolic decompensation that leads to a myopathic response? This needs to be clarified.

Finally, it is worth commenting that in the references the authors rely on a decade old review article as a source of information about the cardiac effects of high fat diet and obesity. A lot of has been unveiled on this topic in the last 10 years, and a more timely and up to date referenced review would benefit both authors and readership as source of information.

Reviewer #3 (Remarks to the Author):

The authors addressed my comments; I still have one concern on how the authors performed the FA - driven mitochondrial respiration. They also need to measure FA-driven respiration in the absence of glutamate, to directly measure e- transfer to the Q pool by fattyacylCoA dehydrogenase. Their current setting allows to measure a mixed response of direct e- transfer as well as TCA cycle, NADH mediated e- transfer at complex I following beta-oxidation.

RESPONSE TO THE REVIEWERS' COMMENTS

First, we would like to thank the reviewers for their constructive comments, which have helped us significantly in improving our manuscript. We have conducted new experiments as suggested by reviewers and have added new data to this revised manuscript. Due to changes in the text and in references, please refer to page number, figure number, reference number in revised manuscript and online data supplement.

Reviewer 1

The revision is improved. While I pointed out several issues in my initial review, I focused on strengthening the data showing that calpain cleaves ERK5 because this is central to mechanism.

The authors first used informatics to identify multiple potential calpain cleavage sites in ERK5. Then, they expressed FLAG-tagged ERK5 in cells, and incubated cell lysates with recombinant calpain in the presence of Ca²⁺. Immunoblotting for FLAG revealed multiple bands smaller than full length FLAG-ERK5 consistent with calpain-mediated cleavage.

I agree that it is CONSISTENT with calpain-mediated cleavage of ERK5 (as are the inhibitor studies presented in the Results), but the problem with this approach is that it does not show that calpain is what is the protease that is actually cleaving ERK5. In theory, calpain could be cleaving and activating another protease, which is then cleaving ERK5.

The most definitive way to show direct cleavage of ERK5 by calpain would have been to use a cell-free system: recombinant ERK5, recombinant calpain, Ca²⁺, and buffer. Then do an immunoblot.

But, in my initial comments to Authors, I said that it would be acceptable to use either an in vitro (cell-free) approach as described above, or a cell-based system. But, if you used a cell-based system, I had assumed that you would try to purify the FLAG-ERK5 by immunoprecipitation before incubating it with the recombinant calpain and Ca²⁺. The way you did it includes the entire cell lysate. An IP step would have lessened concerns that another protease is involved in cleaving ERK5.

But, there is a third option. I would be willing to accept that calpain might be directly cleaving ERK5 if you could (a) show a diagram of the ERK5 protein indicating where the potential calpain cleavage sites are (you should probably include this in Supp Fig 8 anyway because it strengthens your case); and (b) definitively identify some of the cleaved bands on the FLAG immunoblot as corresponding to some of the predicted calpain cleavage fragments based on your informatics analysis.

So, two summarize: three options: (a) the cell-free approach; (b) the approach you used but adding IP: FLAG before incubating with calpain and Ca²⁺; or (c) use with your present data but identify some of your bands as corresponding to predicted calpain cleavage fragments based on your informatics.

Response: We are most grateful to the reviewer for not only pointing out the weakness in our previous experimental design, but also providing practical approaches to address this point. Following the reviewer's suggestion, we have performed two sets of experiments: 1) We have used cell-free system, in which 5pmol or 15pmol recombinant human ERK5 (Abcam, ab126913) and

2 μ g recombinant human calpain-1 (Sigma, C6108) in the absence or presence of 0.25mM Ca²⁺ were incubated for 1 min or 5 mins, and immunoblot was carried out. 2) We have also used cell-based system, in which an immunoprecipitation step was included. We purified the FLAG-ERK5 by immunoprecipitation before incubating with the recombinant human calpain-1 (Sigma, C6108) in the absence or presence of 0.25mM Ca²⁺ for 30 minutes, thereafter subject to immunoblot analysis.

Together, both approaches (cell-free and cell-based) demonstrated that calpain-1 directly cleaves ERK5.

The new data has been added in the revised manuscript.

Result section: page 6, lines 34-41.

New sentences added in to describe these experiments and obtained result.

New data added in **Supplementary Fig. 8** to present aforementioned result, and new figure legend provided.

Methods: page 14, lines 15-25.

Reviewer 2

The authors have responded constructively with new data and importantly improved experimental protocols. A few minor weaknesses do remain, however. The authors rely exclusively on inference in implicating DAG and ceramides as lipotoxic with no direct evidence that either play a role in the dysfunction and pathogenesis of the cardiomyopathy that they observe. A statement should be included to clarify this. I would also contend that many things change, both up and down, in hearts subjected to high fat diet and nutrient overload, all of which are part of a compensatory response to metabolic stress. It would be good if the authors can address the approach, conceptually, because if you take something out of the heart and the heart performs poorly, then you put it back in and it alleviates the dysfunction, it merely proves that all of the components serve a purpose. Does the work anything more than one of many components that regulate the cardiac response to stress?

The abstract states that “Mechanistic studies revealed...” This qualifying term should be left to the readership to determine how mechanistic the study is. Alternative text should replace this opinionated adjective – perhaps “The current study.” or “The protocols revealed...” as suggestions and examples – but, of course, to be determined by the authors.

A question regards the authors’ meaning and intent of the text on page 8, 2nd paragraph, line 16, starting “When the heart works at a compensated stage against metabolic stress.” This is difficult to understand. Are the authors saying something about the actual mechanical work of the heart against a change in afterload? Are they talking instead about a metabolic decompensation that leads to a myopathic response? This needs to be clarified.

Finally, it is worth commenting that in the references the authors rely on a decade old review article as a source of information about the cardiac effects of high fat diet and obesity. A lot of has been unveiled on this topic in the last 10 years, and a more timely and up to date referenced review would benefit both authors and readership as source of information.

Response: We thank the reviewer very much for the positive comment and constructive suggestions. Taking on the reviewer's suggestions, we have made following improvements:

i) A statement is included in discussion section to clarify that although we observed changes in DAG and ceramide species, but we cannot ascertain these lipids are toxic, and future studies are needed (**Discussion, page 8, line 48 to page 9, line 1**). In addition, we have removed "toxic" from description of DAG and ceramides throughout the text.

ii) The reviewer raised a very important conceptual point regarding compensatory responses to metabolic stress in ERK5 models. We have included a paragraph to discuss this issue (**Discussion, page 8, lines 37-47**). We agree that genetically-modified mouse models have limitations in understanding gene functions, however, through refining mouse models by cell/tissue or time-specific modifications, in combination with other technologies/systems, we are able to define whether a gene plays roles during biological events or disease development and progression. Indeed, substantial biological information has been obtained from these mouse models, and we will continue discovering ways to circumvent hurdles in using these models. Nevertheless, genetic modification mouse models remain an invaluable research tool, despite their imperfections. We hope to convince the reviewer and readers that ERK5 plays an important role in protecting mitochondrial function in response to FFA stress, and what we observed in ERK5-CKO and overexpression model is not secondary effect. In addition, those "up and down" genes may act as ERK5 downstream effectors in response to FFA stress, which will subject to future studies.

iii) The abstract states that "Mechanistic studies revealed..." has been changed to "Further studies revealed..."

iv) In discussion section "When the heart works at a compensated stage against metabolic stress..." has been changed to "When the heart is in response to acute metabolic stress..." (**Discussion, page 8, lines 8-9**). We tried to describe a notion of cardiac remodeling in response to metabolic stress over different phases. Facing acute metabolic stress, the heart is in its adaption stage, in which ERK5 remains intact for positive regulation of mitochondrial function, and cardiac contractility is normal. However, when metabolic stress becomes sustained, ERK5 degradation by calpain-1 instigates a cascade of pathological changes reminiscent of metabolic cardiomyopathy, such as mitochondrial dysfunction, energy insufficiency, ROS accumulation, cell death, contractility deteriorated and heart failure.

v) Up-to-date references regarding the cardiac effects of high fat diet and obesity have been replaced as **ref. 3, 6, 7, 13**.

Reviewer 3:

The authors addressed my comments; I still have one concern on how the authors performed the FA-driven mitochondrial respiration. They also need to measure FA-driven respiration in the absence of glutamate, to directly measure e-transfer to the Q pool by fattyacylCoA dehydrogenase. Their current setting allows measuring a mixed response of direct e-transfer as well as TCA cycle; NADH mediated e-transfer at complex I following beta-oxidation.

Response: We thank the reviewer for this comment. In the 1st round of revision, we measured fatty acid driven respiration in mitochondrial preparations using palmitoylcarnitine and malate (not glutamate) as substrates; this was written in method section. Malate was included for the following reason:

In the fatty acid oxidation pathway control state (F, FAO), one or several fatty acids are supplied to feed electrons into the F-junction through fatty Acyl CoA dehydrogenase (reduced form FADH₂), to electron transferring flavoprotein (ETF), and further through the Q-junction to Complex III (CIII). FAO not only depends on electron transfer through the F-junction (which is typically rate-limiting) but simultaneously generates NADH and thus depends on N-junction throughput. Hence FAO can be inhibited completely by inhibition of Complex I (CI). Importantly, FAO generates Acetyl CoA, and the accumulation of this product in mitochondrial preparations will inhibit FAO. For this reason, it is common practice to add malate (in addition to fatty acid substrates) to feed the citric acid cycle and avoid accumulation of Acetyl CoA. Malate is oxidized in a reaction catalyzed by malate dehydrogenase to oxaloacetate (yielding NADH), which then stimulates the entry of Acetyl CoA into the TCA cycle catalyzed by citrate synthase. However, malate alone cannot support respiration in mitochondrial preparations, especially at the low concentration that we applied it (2mM). This is because oxaloacetate cannot be metabolized further in the absence of a source of Acetyl-CoA (e.g. pyruvate, glutamate or fatty acids). Since pyruvate and glutamate were absent from the preparation, any respiration from palmitolcarnitine + malate is considered FAO (rather than other metabolic pathways, such as glycolysis or glutamate metabolism). In the current revised manuscript, we have written fatty acid driven respiration in mitochondrial preparations in a separate paragraph to make this part more clear.

Methods: page 11, lines 35-42.

REVIEWERS' COMMENTS:

Reviewer #1 (Remarks to the Author):

The authors were responsive to my request to demonstrate that Ca²⁺/calpain is able to cleave ERK5 directly (Supp Fig 8a and b; text 6). They actually performed two of the approaches I suggested: (a) using recombinant ERK5 in a cell-free system; and (b) immunoprecipitated ERK5 from heterologous cells transfected with a tagged construct. Both experiments make the point that Ca²⁺-activated calpain can cleave ERK5. There are a few minor aspects about Supp Fig 8a (the cell-free experiment) that I could question such as: (i) why the increasing input of ERK5 going from 5 pmol to 15 pmol in lanes 1 to 2 did not result in more ERK5 on the blot (under non-cleaved, control conditions); and (ii) whether they believe that 5 pmol ERK under cleaved conditions resulted in complete disappearance of the protein (lanes 5 and 8) while the same experiment with 15 pmol ERK5 resulted in a ladder of cleaved fragments (lanes 6 and 9) – they would likely respond that all the protein was degraded with the lower input of substrate. But, all of this is minor as comparison of lane 3 with either lanes 6 or 9 proves the point that Ca²⁺-activated calpain can cleave ERK5 directly. Moreover Fig 8b is completely clear. Taken together with their demonstration that inhibition of calpain-1 inhibits ERK5 cleavage (Fig 6g), their data support the idea that this can happen and be direct. Whether it actually is direct will require mapping and mutation of calpain cleavage sites on ERK5, a large undertaking that I believe would be unfair to request. Thus, the data now in the paper support the central mechanism they propose.

Reviewer #3 (Remarks to the Author):

Authors satisfactorily addressed my concern

RESPONSE TO THE REVIEWER'S COMMENT

First, we would like to thank the reviewer for the constructive comment, which have helped us in improving our manuscript. Due to changes, please refer to figure in revised online supplementary data.

Reviewer #1 (Remarks to the Author):

The authors were responsive to my request to demonstrate that Ca²⁺/calpain is able to cleave ERK5 directly (Supp Fig 8a and b; text 6). They actually performed two of the approaches I suggested: (a) using recombinant ERK5 in a cell-free system; and (b) immunoprecipitated ERK5 from heterologous cells transfected with a tagged construct. Both experiments make the point that Ca²⁺-activated calpain can cleave ERK5. There are a few minor aspects about Supp Fig 8a (the cell-free experiment) that I could question such as: (i) why the increasing input of ERK5 going from 5 pmol to 15 pmol in lanes 1 to 2 did not result in more ERK5 on the blot (under non-cleaved, control conditions); and (ii) whether they believe that 5 pmol ERK under cleaved conditions resulted in complete disappearance of the protein (lanes 5 and 8) while the same experiment with 15 pmol ERK5 resulted in a ladder of cleaved fragments (lanes 6 and 9) – they would likely respond that all the protein was degraded with the lower input of substrate. But, all of this is minor as comparison of lane 3 with either lanes 6 or 9 proves the point that Ca²⁺-activated calpain can cleave ERK5 directly. Moreover Fig 8b is completely clear. Taken together with their demonstration that inhibition of calpain-1 inhibits ERK5 cleavage (Fig 6g), their data support the idea that this can happen and be direct. Whether it actually is direct will require mapping and mutation of calpain cleavage sites on ERK5, a large undertaking that I believe would be unfair to request. Thus, the data now in the paper support the central mechanism they propose.

Response: We are most grateful to the reviewer pointing out the concern. (i) As 15pmol purified recombinant protein is over-concentrated in the loading mixture, which resulted in a quenched signal by the sensitive Amersham ECL Prime Western Blotting Detection Reagent. So we repeated the same experiment with Thermo Scientific Pierce ECL Western Blotting Substrate, and presented the new figure in **Supplementary Fig. 8a**. (ii) As we showed, in the presence of Ca²⁺, less protein was disappeared, while more protein was observed as multiple bands by incubating with Caplain1 protein, hence, we believe Calpain1 is capable of quickly cleaving ERK5 protein by multiple sites (also confirmed with *in silico* sequence screening) to non-functional small fragments which can be degraded eventually (in cells) or are undetectable by the antibody (in the cell free system).